# Reduced ultrafine particle levels in São Paulo's atmosphere during shifts from gasoline to ethanol use

Alberto Salvo[1], Joel Brito (iD)[2,4], Paulo Artaxo[2] & Franz M. Geiger[3]

Despite ethanol's penetration into urban transportation, observational evidence quantifying the consequence for the atmospheric particulate burden during actual, not hypothetical, fuel-fleet shifts, has been lacking. Here we analyze aerosol, meteorological, traffic, and consumer behavior data and find, empirically, that ambient number concentrations of 7–100-nm diameter particles rise by one-third during the morning commute when higher ethanol prices induce 2 million drivers in the real-world megacity of São Paulo to substitute to gasoline use (95% confidence intervals: +4,154 to +13,272 cm$^{-3}$). Similarly, concentrations fall when consumers return to ethanol. Changes in larger particle concentrations, including US-regulated PM2.5, are statistically indistinguishable from zero. The prospect of increased biofuel use and mounting evidence on ultrafines' health effects make our result acutely policy relevant, to be weighed against possible ozone increases. The finding motivates further studies in real-world environments. We innovate in using econometrics to quantify a key source of urban ultrafine particles.

---

[1] Department of Economics, National University of Singapore, 10 Kent Ridge Crescent, Singapore 119260, Singapore. [2] Institute of Physics, University of São Paulo, Rua do Matao, Travessa R, 187, 05508-090, São Paulo, São Paulo, Brazil. [3] Department of Chemistry, Northwestern University, 2145 Sheridan Road, Evanston, IL 60208, USA. [4]Present address: Laboratory for Meteorological Physics (LaMP), Université Clermont Auvergne, Aalto University, Clermont-Ferrand F-63000, France. Correspondence and requests for materials should be addressed to A.S. (email: albertosalvo@nus.edu.sg)

Vehicular emissions are main contributors to urban air pollution within megacities[1]. Of key relevance to both health[2-5] and climate change[6, 7] policies is particulate matter (PM), a broadly defined class of ambient air pollutants[8, 9]. Around the world, gasoline is the typical fuel source for the passenger-car and motorcycle fleets that circulate in urban areas, outnumbering heavy-duty diesel vehicles by an order of magnitude or so. With the introduction of biofuel ethanol to the fuel mix witnessed in countries, such as Brazil, Sweden and the United States, as both a substitute for gasoline and as a fuel additive, it is timely to assess the effect such changes have on ambient particle levels across the size range[10]. Some controlled emissions studies show improved combustion efficiency and lower tailpipe emissions as the ethanol fraction in gasoline increases and, in particular, research in the laboratory indicates that gasoline combustion can lead to larger amounts of ultrafine particles (<100 nm in diameter) when compared to ethanol combustion[11-19]. Beyond the tailpipe and the lab, a few modeling studies have focused on ambient air, attempting to predict the impact on PM2.5 levels (PM up to 2.5 μm in diameter) of the adoption of E20 or E85[20-22], yet such studies are based on hypothetical fuel shifts, and ignore the currently unregulated health-relevant ultrafine range[23-27]. The variation in PM concentrations in ambient air during actual fuel shifts, in the real-world setting of a major metropolitan area undergoing a period of large-scale fluctuations in gasoline vs. ethanol use, has not been assessed, nor has the particle size dependence on the fuel mix been evaluated until now.

The one location that features episodes of large-scale shifts in fuel mix as well as well-maintained monitoring networks for air, weather and road traffic is the subtropical megacity of São Paulo. Urban São Paulo is home to about 20 million people and 6 million passenger cars, with gasoline-ethanol "flex-fuel" vehicles accounting for over half of vehicle miles traveled. Due to significant investment into sugarcane ethanol supply and demand[28], ethanol prices that fluctuate with the world sugar market, and government-controlled gasoline prices, there has been large-scale switching by consumers between ethanol (E100) and gasoline (a E20 blend typically), fuels that are ubiquitous at retail[29, 30].

In this study, we combine aerosol size distribution measurements between 7 and 800 nm and mass concentration measurements for black carbon (BC) and PM2.5 with an econometric approach to evaluate how the gasoline-ethanol fuel mix impacts ambient particle levels in urban São Paulo across a wide range of sizes. The method incorporates consumer responses to price movements at the pump and examines pollutant concentration, meteorology and road traffic observations at the street-hour level. The longest sample period among the data sets that we use is November 1, 2008 to May 31, 2013, excluding the colder months of June to September. This sample period includes two episodes of large variation in ethanol prices and subdued movement in gasoline prices, over the spring to fall 2009–2010 and again over the spring to fall 2010–2011. These large fluctuations in the price of ethanol relative to gasoline, and the induced shifts in consumer choice at the pump between ethanol and gasoline, were driven by developments in world food and energy markets,[31, 32] and not concerns about air quality in São Paulo. Short-run fluctuations in relative fuel prices, while shifting consumers' choice of fuels, did not impact price-inelastic demand for driving or travel behavior[33]. Tunnel studies in the area attribute particle sizes below 100 nm to direct emissions from both light and heavy vehicles[34], and diesel combustion, used exclusively in a fleet of 0.3 million heavy vehicles, was invariant to movements in the ethanol-to-gasoline price ratio. Given this background, we determine how particle concentrations across PM2.5 to ultrafines varied in the real-world setting of São Paulo as the metropolis underwent periods of increased—followed by decreased—gasoline relative to ethanol use, once potential confounding factors are accounted for, including temperature, wind, boundary layer height, precipitation, the spatial distribution of traffic, and even drifts.

A previous application of the econometric method to these price-induced natural experiments addressed only regulated gaseous pollutants routinely measured by the environmental authority[33], compared to the field-derived particle size distribution measurements we now examine[35, 36]. That study found that shifts from gasoline to ethanol use increased ozone concentrations, countering the reduction in ambient number concentrations of <50 nm diameter nanoparticles that we now report. Our "purely empirical"[37] approach provides a concrete benchmark for alternative approaches used to evaluate urban air pollution, specifically those based on emissions inventories, the analysis of exhaust emissions or smog chambers, source apportionment studies and chemical modeling[38, 39]. Motivated by a recent modeling study associating PM with higher mortality when compared to ozone[3], a proposed next step in this research agenda is to evaluate how public health outcomes co-varied with the ethanol fraction relative to gasoline, as ozone rose whereas ultrafines fell while PM2.5 remained invariant.

## Results

**Ultrafines rose with shift to gasoline and fell upon return.** Our two-step multivariate regression model considers price-induced shifts in consumer fuel shares in a first step, and the impact of these consumer choices, gasoline vs. ethanol, on ambient air in a second step[33, 40]. As the first step, we require a consumer demand model[30, 41] to predict day-to-day quantities from day-to-day prices because high-frequency fuel quantity data for the São Paulo metropolis are not available, only daily price data. In the second step, the econometric/statistical approach corrects for[37, 42, 43] potentially high variability in particle levels. Specifically, the analysis fixes or controls for potential factors of nanoparticle variation[8, 10], including the distance of measurement from roads, the time of day, the day of the week, seasonality, longer term trends such as growth and compositional changes in the vehicle fleet, key meteorological variables, traffic congestion, and the combustion of fuels other than gasoline and ethanol, which are our object of interest (Table 1). The econometric approach requires that the analyst give careful consideration to whether remaining, unobservable determinants of nanoparticles might co-vary with the gasoline-ethanol mix, and the evidence suggests not (Methods).

Figure 1 and Table 2 summarize our main results. We both plot and in the table's first row report the estimated changes in ultrafine (7–100-nm diameter), PM100-800 nm, BC, PM2.5, and ozone concentrations scaled for a 50-percentage point shift in the gasoline share in the flex fleet, from 30 to 80%. Induced by the most marked episode of fluctuation in ethanol prices in the past decade, shifts in gasoline use of this magnitude—a rise followed by a fall—were observed from mid-summer to mid-fall of 2011 (Fig. 2a). This was quite a seasonally homogeneous five-month period, for example, with temperatures trending downward only slightly and during which there were no school breaks, noting that large seasonal influences on ambient particles might otherwise be hard to control (correct) for.

Both Fig. 1 and Table 2 report 95% confidence intervals (CI), i.e., with about two standard errors on either side of a point estimate (point estimate ± 1.96 × standard error; see Table 2 notes). Estimated effects from raising the gasoline share on BC mass concentration (reported for 08:00), PM2.5 mass concentration (24-h) and PM100-800 nm number concentration (08:00) are statistically insignificant from zero (Supplementary Notes 1, 2

**Table 1 Description of the different data sets that the present study combines including summary statistics**

| Variable and unit of measurement (and method, if relevant) | Data Source | Full sample period[a] | Sampling sites | Data frequency | No. of observations | Mean | Std. Dev. | Min. | Max. |
|---|---|---|---|---|---|---|---|---|---|
| *Particle pollution variable* | | | | | | | | | |
| PM2.5 mass concentration, 24-h filter ($\mu g\, m^{-3}$) | CETESB | 11/2008–5/2013 | Three[b] | 24-h | 727 | 16.39 | 9.91 | 1.00 | 68.00 |
| PM2.5 mass concentration, beta continuous ($\mu g\, m^{-3}$) | CETESB | 1/2011–5/2013 | Three[c] | 1-h | 43,571 | 20.44 | 14.94 | 0.00 | 160.00 |
| Black carbon (BC) mass concentration, MAAP ($\mu g\, m^{-3}$) | Own | 10/2010–4/2011[d] | One (USP) | 1-h | 6,152 | 3.23 | 2.69 | 0.06 | 15.67 |
| Ultrafine particle number concentration (UFP) 7–100 nm, DMPS ($cm^{-3}$) | Own | 10/2010–9/2011 | One (USP)[e] | 1-h | 6,454 | 14,561 | 6,384 | 1,339 | 56,019 |
| PM 100–800 nm number concentration, DMPS ($cm^{-3}$) | Own | 10/2010–9/2011 | One (USP)[e] | 1-h | 6,454 | 3,161 | 2,900 | 86 | 22,291 |
| *Fuel mix variables (light vehicles and motorcycles)* | | | | | | | | | |
| Ratio of ethanol-to-gasoline regular-grade prices per litre (%) | ANP (at the pump) | 11/2008–5/2013 | Median SPMA[f] | Daily | 1,673 | 0.64 | 0.07 | 0.49 | 0.85 |
| Gasoline share in the flex-fuel light-vehicle fleet (%) | Salvo-Huse (2013) | 11/2008–5/2013 | Estimated[g] | Daily | 1,673 | 0.35 | 0.14 | 0.11 | 0.76 |
| Ethanol share in the flex-fuel light-vehicle fleet (%) | Salvo-Huse (2013) | 11/2008–5/2013 | Estimated[g] | Daily | 1,673 | 0.65 | 0.14 | 0.89 | 0.24 |
| Gasoline share among all gasoline and ethanol consumers (%) | ANP (wholesalers) | 11/2008–5/2013 | SP state[h] | Monthly | 55 | 0.63 | 0.09 | 0.51 | 0.82 |
| Ethanol share among all gasoline and ethanol consumers (%) | ANP (wholesalers) | 11/2008–5/2013 | SP state[h] | Monthly | 55 | 0.37 | 0.09 | 0.49 | 0.18 |
| *Control variables* | | | | | | | | | |
| Solar radiation ($W\, m^{-2}$) | CETESB | 11/2008–5/2013 | Mean SPMA[f] | 1-h | 40,112 | 175.73 | 257.95 | 0.00 | 1280.40 |
| Ground temperature (°C) | CETESB | 11/2008–5/2013 | Mean SPMA[f] | 1-h | 40,144 | 20.77 | 4.84 | 5.53 | 38.40 |
| Relative humidity (%) | CETESB | 11/2008–5/2013 | Mean SPMA[f] | 1-h | 40,013 | 77.28 | 17.73 | 12.30 | 98.90 |
| Wind speed ($m\, s^{-1}$) | CETESB | 11/2008–5/2013 | Mean SPMA[f] | 1-h | 40,145 | 1.37 | 0.76 | 0.00 | 4.36 |
| Wind blows from North–East (yes = 1) | CETESB | 11/2008–5/2013 | SPMA[f,i] | 1-h | 155,308 | 0.15 | 0.35 | 0.00 | 1.00 |
| Wind blows from South–East (yes = 1) | CETESB | 11/2008–5/2013 | SPMA[f,i] | 1-h | 155,308 | 0.41 | 0.49 | 0.00 | 1.00 |
| Wind blows from South–West (yes = 1) | CETESB | 11/2008–5/2013 | SPMA[f,i] | 1-h | 155,308 | 0.10 | 0.30 | 0.00 | 1.00 |
| Wind blows from North–West (yes = 1) | CETESB | 11/2008–5/2013 | SPMA[f,i] | 1-h | 155,308 | 0.17 | 0.38 | 0.00 | 1.00 |
| Precipitation ($mm\, h^{-1}$) | INMET | 11/2008–5/2013 | SPMA[f] | 1-h | 40,085 | 0.21 | 1.57 | 0.00 | 58.40 |
| Thermal inversion at 09:00 with base of layer 0–199 m (yes = 1) | FAB | 11/2008–5/2013 | SPMA[f] | Daily | 1,671 | 0.08 | 0.27 | 0.00 | 1.00 |
| Thermal inversion at 09:00 with base of layer 200–499 m (yes = 1) | FAB | 11/2008–5/2013 | SPMA[f] | Daily | 1,671 | 0.26 | 0.44 | 0.00 | 1.00 |
| Road congestion at the citywide level (km) | CET | 11/2008–5/2013 | SP city[h] | 1-h | 40,152 | 24.65 | 36.57 | 0.00 | 294.66 |
| Road congestion in the North region of SP city (km) | CET | 11/2008–5/2013 | SP city[h] | 1-h | 40,152 | 0.65 | 1.56 | 0.00 | 21.59 |
| Road congestion in the East region of SP city (km) | CET | 11/2008–5/2013 | SP city[h] | 1-h | 40,152 | 6.23 | 10.03 | 0.00 | 99.51 |
| Road congestion in the South region of SP city (km) | CET | 11/2008–5/2013 | SP city[h] | 1-h | 40,152 | 5.15 | 8.44 | 0.00 | 77.55 |
| Road congestion in the West region of SP city (km) | CET | 11/2008–5/2013 | SP city[h] | 1-h | 40,152 | 5.50 | 9.19 | 0.00 | 89.76 |
| Road congestion in the Center region of SP city (km) | CET | 11/2008–5/2013 | SP city[h] | 1-h | 40,152 | 7.11 | 11.03 | 0.00 | 81.35 |
| Number of aircraft departing from Congonhas airport ($h^{-1}$) | ANAC | 11/2008–5/2013 | CGN airport[j] | 1-h | 40,140 | 9.03 | 6.43 | 0.00 | 32.00 |
| Number of aircraft landing at Congonhas airport ($h^{-1}$) | ANAC | 11/2008–5/2013 | CGN airport[j] | 1-h | 40,140 | 9.01 | 6.51 | 0.00 | 29.00 |
| *Diesel prices and usage (heavy vehicles)* | | | | | | | | | |
| Diesel real price index (October 2008 = 100, IPCA) | IBGE | 11/2008–5/2013 | SPMA[f] | Monthly | 55 | 86.06 | 6.36 | 78.25 | 100.85 |
| Ridership on diesel buses in the public transport system ($\times 10^6\, day^{-1}$) | SPTrans | 11/2008–5/2013 | SPMA[f] | Monthly | 55 | 7.96 | 0.44 | 6.78 | 8.65 |

[a]Samples described here include the colder months of June to September and all days of the week
[b]Cerqueira César, Ibirapuera and Pinheiros air monitoring sites
[c]Congonhas, Pinheiros and University of São Paulo/IPEN air monitoring sites
[d]Sampling additionally occurred during 8–11/2012
[e]DMPS data validated against an independent CPC operated concurrently
[f]SPMA denotes São Paulo Metropolitan Area (São Paulo metropolis)
[g]Estimated using actual consumer choices at varying prices
[h]SP denotes São Paulo
[i]Wind monitors at Ibirapuera, Osasco, Pinheiros and Santana stations
[j]CGN denotes Congonhas. See Methods for the data sources beyond the acronyms provided here

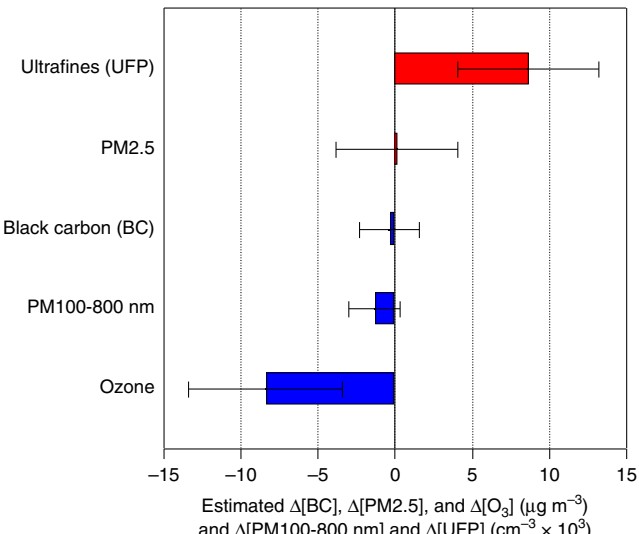

**Fig. 1** Estimated changes in pollutant concentrations. For varying composition, size range, and time-of-day window, in the São Paulo metropolitan area as the gasoline share in the flex-fuel fleet rises from 30 to 80 percentage points. Submicron particles and BC correspond to readings at 08:00, PM2.5 are 24-h means, and ozone are afternoon means between 12:00 and 16:00. Sample periods are January to May 2011 for submicron particles, October 2010 to April 2011 and October to November 2012 for BC, and November 2008 to May 2013 for PM2.5 and ozone. 95% Confidence Intervals (CI) are shown. Source: Specifications reported in Table 2

and 4). Previous research identifies diesel use in heavy vehicles as the main source of BC in the metropolis,[34, 44, 45] and diesel combustion did not fluctuate in tandem with the gasoline-ethanol mix (Supplementary Figs. 5–8). This provides an explanation for the insignificant impact on BC (CI $-0.3 \pm 1.9 \, \mu g \, m^{-3}$) from raising the proportion of the flex fleet burning gasoline rather than ethanol. Figure 3 shows that all estimated gasoline vs. ethanol effects reported in Fig. 1 and Table 2 remain unchanged when we include monthly ridership in diesel buses in the metropolis as an additional control in our regression models (Supplementary Fig. 6).

In contrast to the large size ranges, concentrations of ultrafines during the morning commute show a clear 30% increase with higher gasoline penetration. Taking the largest estimated change over the day, at 08:00, a 30–80% increase in gasoline penetration (equivalent to a 70–20% decrease in ethanol penetration) is associated with a $8,713 \pm 4,559 \, cm^{-3}$ increase in ambient number concentrations of <100 nm diameter nanoparticles, i.e., a CI between +4,154 and +13,272 $cm^{-3}$. The fact that the experimental lever, the share of gasoline, was pulled in both directions—up then down—coinciding with movement in ultrafine levels—up then down—strengthens our result. In particular, the co-variation that we uncover is not estimated off a trend, which our regression models correct for (Table 2), and as such is unlikely to suffer from omitted variable bias.

Also reassuring is the estimated association between meteorology and pollutant concentrations reported in Table 2, such as the negative and statistically significant effect of wind speed on all measured parameters[46]. To illustrate the method, the last column of Table 2 reveals the reduction in afternoon ozone levels

---

**Table 2 Changes to particle and ozone concentrations associated with variation in the gasoline-ethanol fuel mix**

| | (1) | (2) | (3) | (4) | (5) |
|---|---|---|---|---|---|
| Column number: | | | | | |
| Dependent variable: | BC | PM2.5 | PM 100–800 nm | UFP 7–100 nm | Ozone |
| Unit: | $\mu g \, m^{-3}$ | $\mu g \, m^{-3}$ | $cm^{-3}$ | $cm^{-3}$ | $\mu g \, m^{-3}$ |
| Mean over hour window: | 08:00 | 24-h | 08:00 | 08:00 | 12:00–16:00 |
| Sample period: | Oct/2010 to Apr/2011 & Oct to Nov/2012 | Nov/2008 to May/2013 | Jan/2011 to May/2011 | Jan/2011 to May/2011 | Nov/2008 to May/2013 |
| Number of sampling sites: | 1 | 3 | 1 | 1 | 12 |
| Source: | Own | CETESB | Own | Own | CETESB |
| Share of Gasoline E20/E25 in the flex fleet rises from 30 to 80% | $-0.3 \pm 1.9$ | $0.2 \pm 3.9$ | $-1,249 \pm 1,669$ | $8,713 \pm 4,559$ | $-8.3 \pm 5.0$ |
| Equivalently, share of Ethanol E100 in the flex fleet falls from 70 to 20% | | | | | |
| *Control variables (to correct for the influence of other determinants of particles)* | | | | | |
| Site-specific linear trend | Yes | Yes | Yes | Yes | Yes |
| Week-of-year fixed effects | No | Yes | No | No | Yes |
| Day-of-week fixed effects | Yes | Yes | Yes | Yes | Yes |
| Radiation (+100 W $m^{-2}$) | $0.5 \pm 0.7$ | $-0.4 \pm 2.2$ | $4 \pm 825$ | $235 \pm 1,798$ | $4.2 \pm 0.7$ |
| Temperature (+1 ºC) | $0.0 \pm 0.2$ | $1.2 \pm 0.5$ | $236 \pm 235$ | $-847 \pm 836$ | $3.1 \pm 0.4$ |
| Humidity (+10%) | $0.1 \pm 0.7$ | $-1.0 \pm 1.6$ | $349 \pm 721$ | $-1,020 \pm 1,712$ | $-4.9 \pm 1.3$ |
| Wind speed (+1 m $s^{-1}$) | $-3.2 \pm 1.2$ | $-6.5 \pm 2.8$ | $-2,102 \pm 1,410$ | $-4,217 \pm 3,489$ | $-13.2 \pm 2.1$ |
| Other meteorological and road traffic conditions (see notes) | Yes | Yes | Yes | Yes | Yes |
| $R^2$ | 62.0% | 73.4% | 76.0% | 69.8% | 70.7% |
| Number of observations | 129 | 511 | 80 | 80 | 13,203 |
| Number of regressors | 18 | 74 | 19 | 19 | 96 |
| Mean value of dependent variable | 6.0 | 13.8 | 3,577 | 18,659 | 72.2 |

Coefficients and 95% confidence intervals, i.e., point estimate ± 2 standard errors. An observation is a date (columns 1, 3, 4) or a date-site pair (columns 2, 5). Samples exclude the colder months of June to September, and include all days of the week (columns 2, 5) or non-holiday weekdays only (columns 1, 3, 4). Radiation, temperature, humidity, and wind speed in the recorded unit. All columns additionally include several precipitation, thermal inversion and road traffic congestion indicators. Columns 1 to 4 further control for wind direction and column 5 follows Supplementary Table 4. Since the longer samples encompass 2010, columns 2, 5 include site-specific intercepts indicating the opening of the Greater São Paulo beltway's southern section on March 31, 2010. The effect of raising the gasoline share in the flex fleet is scaled for in-sample variation from 30 to 80%. The corresponding variation in the ethanol share is one minus variation in the gasoline share. Ordinary Least Squares (OLS) estimates, with standard errors calculated by bootstrapping (200 samples each): (i) the consumer-level fuel choice data, to account for sampling variation in the predicted gasoline share in a first-step consumer demand model, and (ii) the pollutant-meterology-traffic data in the second-step particle regression, clustering by date

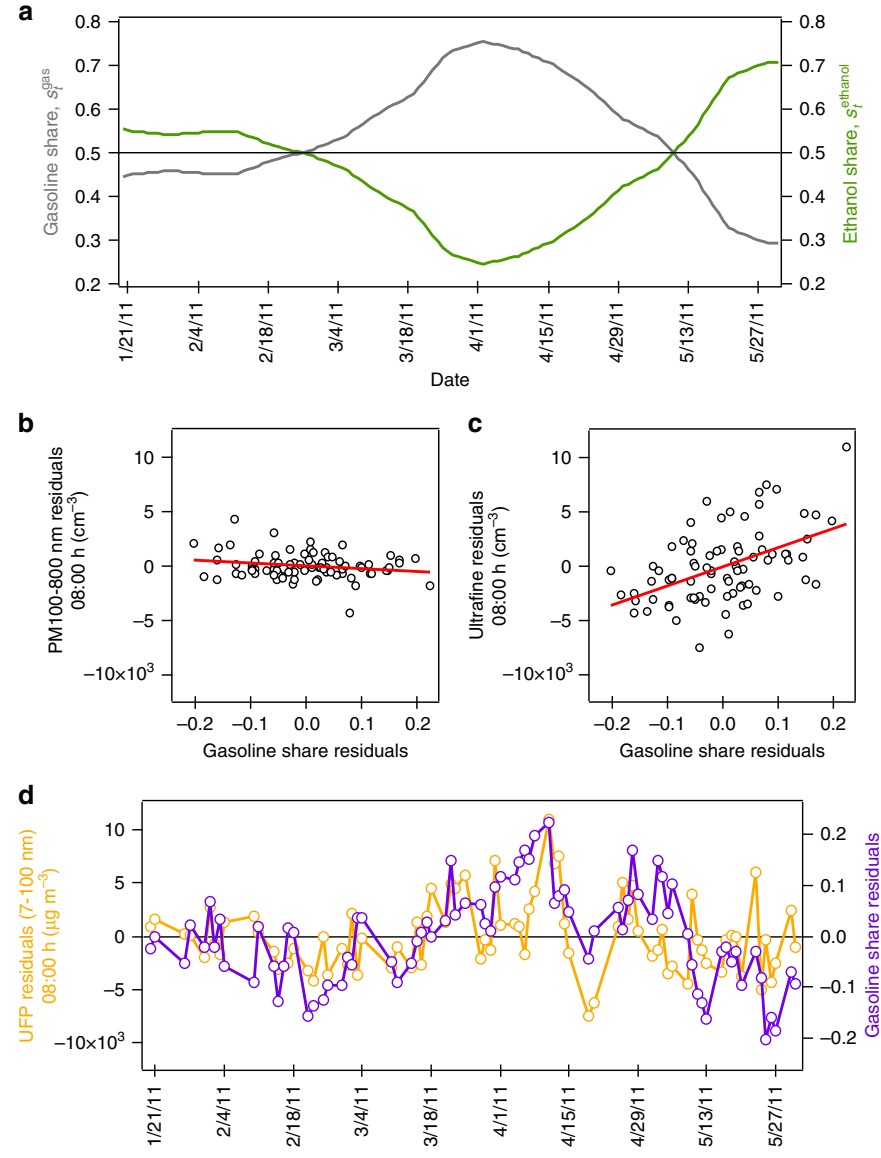

**Fig. 2** Submicron particles and the gasoline share. **a** Fuel share variation among flex-fuel vehicles from January to May 2011. **b** Co-variation of PM 100–800 nm and **c** ultrafine number concentration residuals with gasoline share residuals for the weekday morning hour of 08:00 in the same period. The *red line* marks the best linear predictor. **d** Morning-hour variation of ultrafine number concentration and gasoline share residuals over the period. Source: Specifications reported in Table 2

previously reported for shifting ethanol to gasoline use[33], consistent with a hydrocarbon-limited regime[37], but estimated here from a longer sample than previously, namely 2008 to 2013, rather than to 2011 (Supplementary Note 3).

**Tight co-variation during morning rush hour**. Figure 2b and c provide a graphical representation of these submicron results. We separately plot residual concentrations at 08:00 for PM100-800 nm and for ultrafines, obtained after filtering out all co-variation with control variables, against the residual gasoline share, obtained in the same way. For clarity, from each raw data series—the 7–100 nm values at 08:00, the 100–800 nm values at 08:00, and the gasoline share—we "partial out" (correct for) any co-variation with observed meteorological and road traffic conditions, as well as systematic day-of-the-week (e.g., Monday vs. Friday) and trending variation (Supplementary Note 6). The illustration considers the two-step model specification reported in Table 2, columns (3) and (4), for a sample restricted to non-

holiday weekdays between January 20 and May 31, 2011, and wind direction and a linear trend added to the vector of controls. We choose this as our preferred specification with a view to limiting unobserved determinants of the particle size distribution, which might bias our estimates of the effect of the fuel mix or make them less precise (Supplementary Note 5 reports robustness to these choices and Methods provides an overview of all estimated regression specifications).

Indeed, the panels show the strong positive association with gasoline for 7–100 nm (Fig. 2c), but not for 100–800 nm (Fig. 2b), at 08:00. We repeat the exercise for 7–100 nm and 100–800 nm measurements in the evening, at 18:00 (Supplementary Fig. 17f,g), and there is no clear relationship.

Figure 2d plots the strikingly tight and statistically significant day-to-day correlation at 08:00: the ultrafine particle levels we uncover, after correcting for the influence of other observed factors, move in lockstep with the gasoline share. In the context of a regression equation, both the outcome variable, ultrafines, and

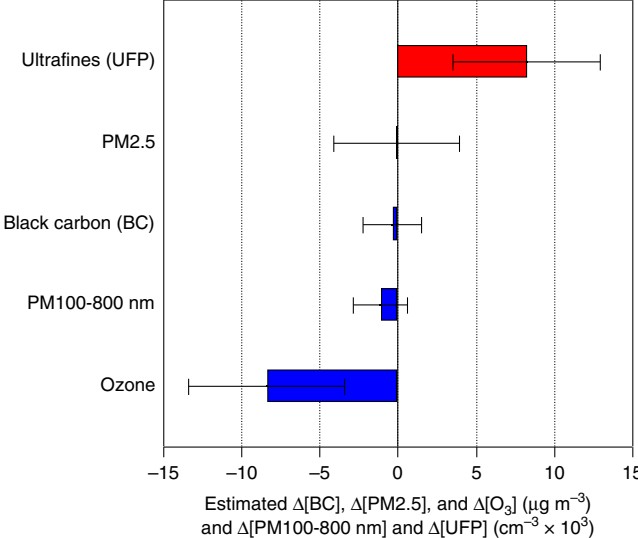

**Fig. 3** Sensitivity to diesel control for changes in pollutant concentrations. For varying composition, size range, and time-of-day window, in the São Paulo metropolitan area as the gasoline share in the flex-fuel fleet rises from 30 to 80 percentage points. Submicron particles and BC correspond to readings at 08:00, PM2.5 are 24-h means, and ozone are afternoon means between 12:00 and 16:00. Sample periods are January to May 2011 for submicron particles, October 2010 to April 2011 and October to November 2012 for BC, and November 2008 to May 2013 for PM2.5 and ozone. 95% CI are shown. Source: Specifications reported in Table 2 additionally controlling for monthly diesel bus ridership in the metropolis' public transportation system (Supplementary Fig. 6)

the key regressor of interest, the gasoline share (which is assumed orthogonal to any remaining unobserved determinants of ultrafines) together move up until the beginning of April, and down thereafter. To emphasize, the result not only accounts for changes in observable meteorological parameters—for example, average daily minimum temperatures in this sample varied from 21 °C in January to 15 °C by May—but also for an unobservable seasonal trend (which should be mild in this sample).

The increase in ultrafine particle concentrations due to increased gasoline penetration manifests itself most clearly in the early morning commute. Figure 4a shows that once corrected for other factors, the positive association between the remaining (residual) concentration in ultrafines and gasoline use is most significant during the morning rush hour. This relationship is not significant during the evening rush hour (though, given a CI of −3,956 to +5,858 $cm^{-3}$ at 18:00, a positive effect cannot be statistically rejected either). Figure 4b shows that changes in PM100-800 nm concentrations with the gasoline share remain indistinguishable from zero during the course of the day.

**Sub-50 nm ultrafines vary most.** To further zoom into which sizes in the ultrafine mode contribute the most to the increased PM concentrations, we integrated the aerosol number concentrations over size bins having increasing width, starting at 7 nm and going up to 800 nm. Figure 4c shows that the most important contributor to the increased (resp., decreased) particle concentration in the ultrafine mode that coincides with the observed increase (resp., decrease) in the gasoline share is the bin of particles having diameters up to 50 nm. No detectable change occurs in the particle concentration beyond that diameter, even all the way up to 800 nm. For comparison, Fig. 4c also shows the 100–800 nm mode, whose number concentration change with fuel mix variation is again indistinguishable from zero. In sum, changes in PM concentrations that coincide with consumers

transitioning from ethanol into gasoline and back to ethanol are not driven by the accumulation mode but instead by the nucleation mode, specifically nanoparticles having diameters <50 nm.

**Variation in 24-h means.** Table 3 reports on regressions that examine variation in 24-h means for: the contribution of nucleation (10–50 nm), Aitken (30–120 nm) and accumulation (70–280 nm) modes to the aerosol particle size distribution[47]; BC mass concentrations; and PM2.5 mass concentrations. We provide two-step model estimates in the odd-numbered columns and, for sensitivity analysis in the even-number columns, estimates from an alternative model based on two-stage least squares (2SLS), with the ethanol-to-gasoline price ratio serving as an "instrumental variable (IV)" for the predicted gasoline share (Methods). Moreover, compared to Table 2, the submicron particle regressions that we report on in Table 3 use samples that are longer, starting October 2010 rather than January 2011, and contain all days of the week, including weekends and holidays. To control for the additional seasonal and weekly variability, we include quarter-of-year and additional day-of-week fixed effects (these allow fitted particle levels to vary systematically by quarter, on public holidays, on Saturdays, etc). PM2.5 levels, routinely monitored by the environmental authority[48] rather than our field campaign, are over multiple years, so we can add more granular week-of-year fixed effects, as these do not subsume the temporal source of fuel mix variation that is our main variable of interest. Two-step model estimates for PM2.5 in column (9) are as in Table 2, column (2).

Across the two model variants (two-step or 2SLS), we obtain a statistically significant and positive association between the gasoline share and the 24-h average contribution of nucleation mode particles, i.e., a 2,794 ± 1,456 $cm^{-3}$ increase, or a 95% CI, between +1,338 and +4,250 $cm^{-3}$ as gasoline usage in the flex fleet rises from 30 to 80% (Table 3, column (1)). This association is consistent with the results over the day and across the size range presented earlier in Fig. 4. Also consistent with the preceding analysis, we do not detect significant associations between gasoline and particle number in the Aitken and accumulation modes (columns (3) to (6)). Consistent with Table 2, the gasoline share is not significantly associated with 24-h BC concentrations in the all-day sample used in Table 3 (columns (7) and (8)). Two-step model estimates are very similar to 2SLS estimates; for example, compare 24-h PM2.5 concentrations in columns (9) and (10). In sum, the findings presented in Table 3 are consistent with those in Table 2.

**Discussion**

The empirical pattern that emerges is characterized by a positive and significant association between the gasoline share and 7–100 nm particle levels in the peak hours of morning travel, and the absence of a statistically significant relationship outside this size range and time window. We repeat the several aspects that give us confidence that our findings are not mere statistical artifacts. First, we infer "differences in differences": differential results for different particle size ranges, namely the nucleation vs. the Aitken and accumulation modes, and the 7–100 nm vs. 100–800 nm size ranges particularly during the morning commute (the first difference is the co-variation with the gasoline share—up and down in tandem over time). Second, in the shorter sample—a period in which meteorology varies mildly and "monotonically" as mid-summer conditions evolve into those that characterize mid-fall—ultrafine particle levels after correcting for confounders move in lockstep with the gasoline share: nanoparticles and gasoline jointly rise until the start of April, then jointly fall through the

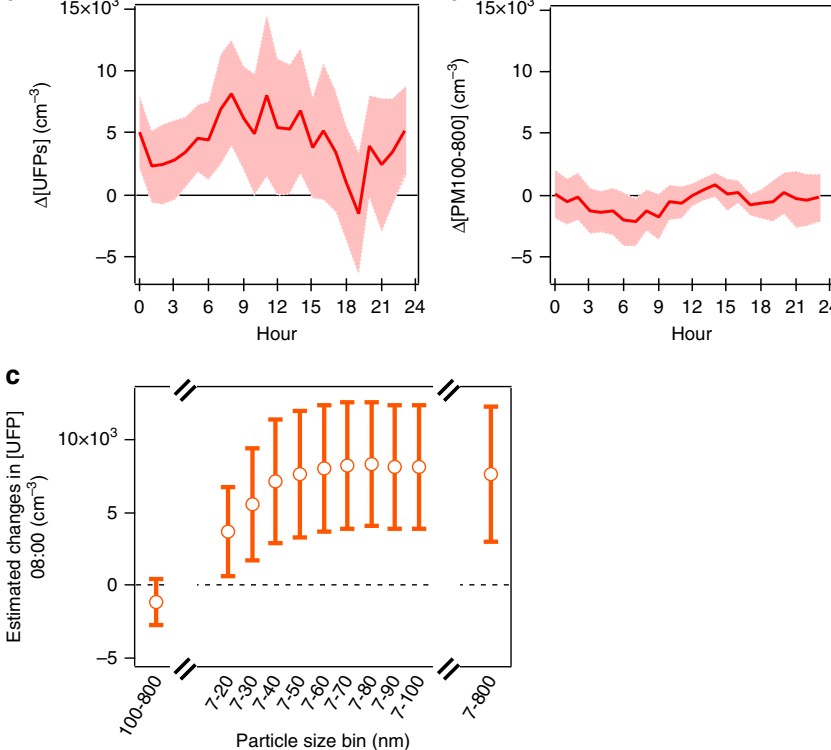

**Fig. 4** Estimated changes over weekday diurnal cycle. **a** 7–100 nm and **b** 100–800 nm particle concentration levels, over the weekday diurnal cycle, associated with a 50-percentage-point rise in gasoline use in the flex-fuel fleet, from 30 to 80%. For clarity, for every hour of the day we plot the 95% CI for the gasoline share's association with the 7–100 nm size range **a**, and the 95% CI for the gasoline share's association with the 100–800 nm size range **b**. **c** Opening the 7–20 nm size bin towards 800 nm, and comparison to the 100–800 nm bin, for the weekday morning hour of 08:00. For clarity, for 08:00 we plot the CI for the gasoline share's association with every size bin. Source: Specifications reported in Supplementary Table 8 **B**, **D**, with sample period restricted to the summer/fall months of January to May 2011 and trend included as seasonality control (same specifications as Table 2 for 7–20 nm and 100–800 nm at 08:00)

end of May. The potential confounding factors that we control for include a trend and meteorological and road traffic conditions recorded concurrent to the day and hour. A third factor that strengthens our findings is that they are consistent with controlled emissions studies and laboratory experiments[13–16, 18]. Fourth, our approach indicates an insignificant association between the light-vehicle gasoline-ethanol mix and BC levels, which are influenced mainly by diesel combustion in heavy vehicles[39, 43, 44]. Fifth, controlling for monthly ridership of diesel buses in the metropolis does not change our estimates, due to bus ridership not varying over the sample period.

We provide the following possible rationalization of the associations identified here. Changes in the <50 nm diameter nanoparticle concentrations in ambient air are consistent with flame combustion experiments, which show emissions of nucleation mode particles decrease with increasing ethanol fraction in gasoline blends from E0, E20, and E50 to E85[13]. Insofar as these laboratory studies are applicable to São Paulo's urban air chemistry, it may be plausible to attribute the reported fluctuations in <50 nm diameter particles during a seasonally similar period to differences in the composition of direct emissions that occurred in tandem. Replacing gasoline-rich fuel blends with ones rich in ethanol may then result in significant reductions in ultrafine—specifically <50 nm diameter nanoparticle—levels, as we indeed estimated from the field-derived size distribution measurements. The chemical analysis of the nanoparticles, which did not occur during the period of fuel switching we studied, would be an important next step towards understanding how gasoline-ethanol mixes impact particle pollution in urban air.

In conclusion, we have combined aerosol, meteorological, traffic, and consumer behavior data in an econometric approach that identifies a statistically significant inverse association between ethanol content in gasoline and ambient <50 nm diameter nanoparticle concentrations. Specifically, we find decreases of up to 25–30% during morning rush hours associated with an in-sample 20 to 70% increase in ethanol penetration (equivalently, 80 to 30% decrease in gasoline use) in the São Paulo flex-fuel fleet. Whether subsequent atmospheric processing and/or secondary material formation are materially influenced by shifts in the fuel mix, and were not captured by our empirical model, is unknown and motivates further studies. As with any empirical observational study, confidence in its findings can only grow as new samples, in space and time, become available, supported by the results from different approaches and analysis techniques. Nevertheless, our result that, after correcting for other influences on particles, higher-followed-by-lower gasoline vs. ethanol use in São Paulo coincided with higher-followed-by-lower <50 nm diameter nanoparticle levels points towards the possibility that the use of ethanol-rich gasoline blends as a transportation fuel may decrease the atmospheric burden of health-relevant ultrafine particles, specifically those that can reach deep into the pulmonary system. This novel result, obtained in the field, is particularly timely as several countries now consider implementing their intended nationally determined contributions to reduce fossil fuel emissions, as agreed at the recent COP-21 in Paris, by increasing biofuel use. Yet, we caution that this environmentally desirable outcome is countered by the increases in local ozone concentrations reported on earlier[33, 49].

**Table 3 Changes to 24-h mean particle concentrations associated with variation in the gasoline-ethanol fuel mix**

| Column number: | (1) | (2) | (3) | (4) | (5) | (6) | (7) | (8) | (9) | (10) |
|---|---|---|---|---|---|---|---|---|---|---|
| Dependent variable: | Nucleation | | Aitken | | Accumulation | | BC | | PM2.5 | |
| Unit: | dN/dlogDp, cm$^{-3}$ | | dN/dlogDp, cm$^{-3}$ | | dN/dlogDp, cm$^{-3}$ | | µg m$^{-3}$ | | µg m$^{-3}$ | |
| Mean over hour window: | 24-h | | 24-h | | 24-h | | 24-h | | 24-h | |
| Sample period: | Oct/2010 to May/2011 | | Oct/2010 to May/2011 | | Oct/2010 to May/2011 | | Oct/2010 to Apr/2011 and Oct to Nov/2012 | | Nov/2008 to May/2013 | |
| Number of sampling sites: | 1 | | 1 | | 1 | | 1 | | 3 | |
| Source: | Own | | Own | | Own | | Own | | CETESB | |
| Estimation: | 2-step model | 2SLS model | 2-step model | 2SLS model | 2-step model | 2SLS model | 2-step model | 2SLS model | 2-step model | 2SLS model |
| Flex fuel share of Gasoline E20/E25 rises from 30 to 80% Equivalently, share of Ethanol E100 falls from 70 to 20% | 2,794 ± 1,456 | 2,783 ± 1,433 | 332 ± 818 | 361 ± 818 | 565 ± 785 | 553 ± 806 | 1.1 ± 1.3 | 1.0 ± 1.2 | 0.2 ± 3.9 | −0.2 ± 2.9 |
| *Control variables (to correct for the influence of other determinants of particles)* | | | | | | | | | | |
| Site-specific linear trend | Yes | Yes | Yes | Yes | Yes | Yes | Yes | Yes | Yes | Yes |
| Quarter-of-year fixed effects | Yes | Yes | Yes | Yes | Yes | Yes | Yes | Yes | – | – |
| Week-of-year fixed effects | – | – | – | – | – | – | – | – | Yes | Yes |
| Day-of-week fixed effects | Yes | Yes | Yes | Yes | Yes | Yes | Yes | Yes | Yes | Yes |
| Radiation (+100 W m$^{-2}$) | −234 ± 664 | −235 ± 607 | 110 ± 386 | 111 ± 361 | 108 ± 259 | 108 ± 226 | 0.0 ± 0.3 | 0.0 ± 0.2 | −0.4 ± 2.2 | −0.3 ± 1.3 |
| Temperature (+1 ºC) | −499 ± 234 | −498 ± 211 | −61 ± 119 | −63 ± 105 | 32 ± 100 | 32 ± 91 | 0.1 ± 0.1 | 0.1 ± 0.1 | 1.2 ± 0.5 | 1.2 ± 0.4 |
| Humidity (+10%) | −1,454 ± 699 | −1,453 ± 543 | −715 ± 384 | −718 ± 322 | −97 ± 265 | −96 ± 243 | −0.4 ± 0.2 | −0.4 ± 0.2 | −1.0 ± 1.6 | −1.0 ± 1.2 |
| Wind speed (+1 m s$^{-1}$) | −569 ± 1,144 | −567 ± 1,046 | −1,910 ± 686 | −1,913 ± 593 | −485 ± 449 | −484 ± 389 | −1.6 ± 0.6 | −1.6 ± 0.5 | −6.5 ± 2.8 | −6.5 ± 2 |
| Other meteorolog. and road traffic conditions (see notes) | Yes | Yes | Yes | Yes | Yes | Yes | Yes | Yes | Yes | Yes |
| $R^2$ | 57.3% | 57.3% | 51.5% | 51.5% | 54.5% | 54.5% | 74.1% | 74.1% | 73.4% | 73.4% |
| Number of observations | 198 | 198 | 198 | 198 | 198 | 198 | 228 | 228 | 511 | 511 |
| Number of regressors | 30 | 30 | 30 | 30 | 30 | 30 | 29 | 29 | 74 | 74 |
| Mean value of dependent variable | 8,755 | 8,755 | 3,320 | 3,320 | 1,494 | 1,494 | 3.3 | 3.3 | 13.8 | 13.8 |

Coefficients and 95% confidence intervals, i.e., point estimate ± 2 standard errors. An observation is a date (columns 1–8) or a date-site pair (columns 9–10). Samples exclude the colder months of June to September and include all days of the week. Radiation, temperature, humidity, and wind speed in the recorded unit. All columns additionally include several wind direction, precipitation, thermal inversion and road traffic congestion indicators. Since the longer sample encompasses 2010, columns 9–10 include site-specific intercepts indicating the opening of the Greater São Paulo beltway's southern section on March 31, 2010. The effect of raising the gasoline share in the flex fleet is scaled for in-sample variation from 30 to 80%. The corresponding variation in the ethanol share is one minus variation in the gasoline share. Ordinary Least Squares estimates in the odd-numbered columns, with standard errors calculated by bootstrapping (200 samples each): (i) the consumer-level fuel choice data, to account for sampling variation in the predicted gasoline share in a first-step consumer demand model, and (ii) the pollutant-meterology-traffic data in the second-step particle regression, clustering by date. Two-Stage Least Squares estimates in the even-numbered columns, with the median ethanol-to-gasoline price ratio across pumping stations instrumenting for the predicted gasoline share in the particle regression equation

## Methods

**Particle sampling methods and coverage**. We combine data from different sources. Among the particle pollution outcomes that we analyze, our most established data set—in terms of both method and temporal coverage—consists of 24-h filter measurements of PM2.5 mass concentration maintained by the environmental authority of the state of São Paulo (CETESB)[48]. Three stationary sites, at varying distance from roads, were sampled with a dichotomous sampler every six days in the city of São Paulo (Supplementary Fig. 1 showing Cerqueira César, Ibirapuera and Pinheiros sites). The sample period we study is November 1, 2008 to May 31, 2013.

This sample period includes two episodes of large variation in ethanol prices and subdued movement in gasoline prices, over the spring to fall 2009–2010 and again over the spring to fall 2010–2011. These large fluctuations in the price of ethanol relative to gasoline, and the induced shift in consumer choice at the pump between ethanol and gasoline, were driven by supply-side shocks, such as a poor sugarcane harvest in India in late 2009. In particular, the pronounced variation in relative ethanol prices was unrelated to the strength of consumer demand for driving or commuting in the São Paulo metropolis, which might otherwise confound our inference of the effect of the fuel mix on air quality. Shifting between gasoline and ethanol, consumers were merely responding to—not causing—the relative price movement at the pump. We return to these consumer shifts below.

A second sample on particles that we examine, also provided by the environmental authority[48], consists of hourly measurements of PM2.5 mass concentration (beta continuous analyzer, model 5014i, Thermo Scientific, Franklin, MA, USA). This higher-frequency (hourly) PM2.5 sampling started in January 2011 at the Congonhas site and continued through the end of our sample period in May 2013. This sample period includes the second and more pronounced of the two episodes of large variation in ethanol prices observed across the metropolis between 2008 and 2013. The Congonhas site lies near an inner city airport, Congonhas airport, and a busy multilane road. Subsequent to fall 2011, the environmental authority began collecting hourly measurements of PM2.5 mass concentration at the IPEN-USP site, starting in August 2011, and at the Pinheiros site, starting in January 2012 (Supplementary Fig. 1). The IPEN-USP site lies inside the Armando Salles de Oliveira campus of the University of São Paulo (USP), in very close proximity to the sampling site for our third data set, described next.

A third data set consists of aerosol particle size distributions in the 7–800 nm range using a Differential mobility particle sizer (DMPS)[40, 41], and BC mass concentrations measured using a Multi angle absorption photometer (MAAP,

model 5012, Thermo Scientific, Franklin, MA, USA). This third sample is not routinely available from an official authority but was collected as part of a field campaign. The campaign lasted from October 2010 to September 2011 (DMPS), and October 2010 to April 2011 followed by August 2012 to November 2012 (MAAP). Importantly, sampling included a seasonally similar period of 4.5 months —from January 20 to May 31, 2011—of marked increase followed by decrease in ethanol prices. This subsample gives us heightened confidence in our submicron particle results.

We validated the DMPS measurements against an independently operated condensation particle counter (CPC, model 3022, TSI Inc., St. Paul, MN, USA), operated concurrently to the DMPS and at a similar lower size cut. As a result of the data validation, less than 2% of the original DMPS data were removed due to a deviation of the integrated aerosol number concentration 50% or higher than the aerosol number concentration measured independently by the CPC. A linear fit between the DMPS integrated concentration and the CPC concentration yields an $R^2$ of 0.99, with a slope of 1.18 (Supplementary Fig. 2a). The diurnal variation (median) and variance (interquartile range) of both measurements show very tight correlation, without differential trends throughout the day (Supplementary Fig. 2b).

Aerosol and sheath flow for the DMPS, CPC setup against an electrometer, compensation for system diffusion losses, and all other calibrations, adjustments and maintenance procedures follow previously published work[50] exactly, and can be found there. All aerosol size distribution measurements were performed with the DMPS with a high time resolution of 10 min for a full measurement cycle. DMPS aerosol particle size distributions were fitted for three lognormal modes[47], allowing the contribution of nucleation (10–50 nm), Aitken (30–120 nm) and accumulation (70–280 nm) modes to be analyzed separately (Supplementary Table 5). Measured parameters were averaged into 1-h data[50].

The DMPS, CPC and MAAP instrumentation was deployed at the roof of a four-storey building located inside the USP Armando Salles de Oliveira campus, about 10 km from the city center in a highly populated area (Supplementary Fig. 1)[50]. This site lies in relative proximity—on the opposite side of the Pinheiros river—to the Pinheiros site, where the environmental authority collected the 24-h PM2.5 samples over a common period. Below we examine the association between BC and PM2.5 mass concentrations measured separately on the same dates in the nearby sites.

The USP campus location for the DMPS/CPC instrumentation very likely provides a lower ultrafine particle number concentration when compared to a

**Table 4 Overview of the estimated regression model specifications**

| Estimates reported in | Dependent variable(s) (and data source) | Sample description[a] | Time aggregation | Fuel mix variable | Estimation procedure (s) | Other sensitivity analysis provided |
|---|---|---|---|---|---|---|
| Supplementary Table 1 | PM2.5 mass concentration, 24-h filter (CETESB) | 11/2008–5/2013, 3 sites every 6 days | 24-h mean | Gasoline share in the flex fleet | OLS + bootstrap | Across the columns, more and alternative controls are introduced, e.g., trend and meteorology |
| Supplementary Table 2 (column 1 in Supplementary Fig. 10) | PM2.5 mass concentration, 24-h filter (CETESB) | 11/2008–5/2013, 3 sites every 6 days | 24-h mean | Gasoline share in the flex fleet, or aggregate fleet | OLS + bootstrap; 2SLS; OLS | Across the columns, the gasoline share and the estimation procedure are varied. Models include controls |
| →Tables 2 and 3 and Fig. 1 report the effect of raising the gasoline share on 24-h PM2.5 levels (model with wind direction controls). Figure 3 reports robustness to diesel control | | | | | | |
| Supplementary Table 3 (column 2 in Supplementary Fig. 10) | PM2.5 mass concentration, beta continuous (CETESB) | 1/2011–5/2013, 3 sites by 2012 | 5-h mean, 07:00 to 11:00 | Gasoline share in the flex fleet, or aggregate fleet | OLS + bootstrap; 2SLS; OLS | Across the columns, the gasoline share and the estimation procedure are varied. Models include controls |
| Supplementary Table 4 (column 3 in Supplementary Fig. 10) | Ozone mass concentration (CETESB) | 11/2008–5/2013, 12 sites | 5-h mean, 12:00 to 16:00 | Gasoline share in the flex fleet, or aggregate fleet | OLS + bootstrap; 2SLS; OLS | Across the columns, the gasoline share and the estimation procedure are varied. Models include controls. Figure 3 reports robustness to diesel control |
| →Table 2 and Fig. 1 report the effect of raising the gasoline share on ozone levels in the early afternoon (reproduced in column 1 of Supplementary Table 4) | | | | | | |
| Supplementary Table 5 | Particle count, nucleation, Aitken, accumulation, BC mass concentration (Own) | 10/2010–5/2011 for DMPS, 1 site (similar periods for other param.) | 24-h mean | Gasoline share in the flex fleet | OLS + bootstrap; 2SLS | Across the columns, the dependent variable and the estimation procedure are varied. Models include controls |
| →Table 3 reports the effect of raising the gasoline share on 24-h nucleation, Aitken, accumulation, BC levels (model with wind direction controls) | | | | | | |
| Supplementary Table 6 (Supplementary Fig. 13 shows panel D) | UFP 7-100 nm (Own) | 10/2010–5/2011, i.e., full period of field campaign, 1 site | 1-h mean[b] | Gasoline share in the flex fleet | OLS + bootstrap; 2SLS | The panels show variation in the sample (all days of the week vs. weekdays only), the estimation procedure, and the effect of wind direction controls |
| Supplementary Table 7 (Supplementary Fig. 13 shows panel D) | PM 100-800 nm (Own) | 10/2010–5/2011, i.e., full period of field campaign, 1 site | 1-h mean[b] | Gasoline share in the flex fleet | OLS + bootstrap; 2SLS | The panels show variation in the sample (all days of the week vs. weekdays only), the estimation procedure, and the effect of wind direction controls |
| Supplementary Table 8 (Supplementary Fig. 14 shows panels B,D) | UFP 7-100 nm and PM 100-800 nm (Own) | 1/2011–5/2011, i.e., more seasonally homogeneous sample | 1-h mean[b] | Gasoline share in the flex fleet | OLS + bootstrap | The panels show variation to including a linear trend vs. not allowing a trend (Specifications otherwise follow panel D, Supplementary Tables 6 & 7.[c]) |
| →Table 2 and Figs. 1 & 2 report the effect of raising the gasoline share on UFP 7-100 nm and PM 100–800 nm levels at 08:00 (reproduced in panels B and D of Supplementary Table 8). →Figure 3 reports robustness to diesel control. Figure 4 shows effects over the day and across the size range using the specification in panels B and D of Supplementary Table 8. | | | | | | |
| Supplementary Figs. 11 & 12 | Particle count and BC mass concentration (Own) | 11/2010–5/2011 for CPC, 1 site (similar period for BC) | 1-h mean[b] | Gasoline share in the flex fleet | OLS + bootstrap | |
| →Table 2 and Fig. 1 report the effect of raising the gasoline share on BC levels at 08:00. Figure 3 reports robustness to diesel control. | | | | | | |
| Supplementary Fig. 9 | PM2.5 mass concentration, beta continuous (CETESB) | 1/2011–5/2013, 3 sites by 2012 | 1-h mean[b] | Gasoline share in the flex fleet | OLS + bootstrap | |

[a]All estimated samples exclude the colder months of June to September
[b]Hour-by-hour regressions
[c]Sample restricted to non-holiday weekdays, wind direction controlled for

roadside or on-road location, suggesting that the absolute changes in particle number concentrations that coincide with the fuel shifts may be different if they were to be measured near road traffic or at the vehicle exhaust[51, 52]. The site has been described as "ideal for tracking ambient aerosols" and "representative of the ambient pollution burden of the city," due to the well-mixed air masses arriving at the location and limited influence of local sources[50]. With 6 million passenger cars and 0.3 million heavy-duty vehicles circulating and emitting across the metropolis, it was critical to choose a site that is representative for the whole urban area. The site is not influenced by one passing accelerating smoky vehicle or idiosyncratic construction next door, nor is it a background site. Importantly, the site lies at a 1 km radius from a major road corridor (Marginal Pinheiros, running northwest to south, spanning over 200 degrees, and 20 express/local lanes in two directions) and is surrounded by busy roads (e.g., Corifeu de Azevedo Marques to the west–southwest). Non-anthropogenic influences, particularly in the ultrafine range, are limited. For example, vegetation on campus is limited compared to the dense vehicular traffic flows that surround it.

In sum, we have access to 24-h PM2.5 mass concentrations measured at three sites every six days between 2008 and 2013; 1-h PM2.5 mass concentrations measured continuously at (essentially) a fourth site between 2011 and 2013; and

parameters in the submicron mode measured continuously at a fifth site for almost one year to September 2011.

**Spatial and temporal variation in PM2.5 and BC levels**. We compared PM2.5 mass concentrations measured on common dates across the different sampling locations and methods (Supplementary Fig. 3a–e). Measurements available at hourly frequency were aggregated into a 24-h mean within each date of measurement, from hour 0:00 to 23:00. Particle levels across sites and methods are highly correlated over time. Similarly, we compared BC to PM2.5 mass concentrations measured in nearby locations on common dates (Supplementary Fig. 3f,g).

There is large variation in particle concentrations between dates—due, for example, to time-varying meteorological conditions, which affect all sites in the same direction—and less variation across site locations. For example, low wind speeds and the occurrence of thermal inversions in the metropolis' atmosphere drive up particle levels measured at all sites. This suggests that if the heavy PM2.5 is well mixed in the atmosphere, then this may be the case for the much lighter ultrafines, and the chosen monitoring site is indeed likely representative of a much

wider area. Moreover, if large fluctuations in ultrafine particle levels were observed at the site for idiosyncratic and unobservable reasons, other than due to determinants such as meteorology and seasonality that we explicitly account for, this would be reflected in large confidence intervals on our estimated effects, which would prevent us from making statistical inference.

In panel a, 24-h filter measurements at both Ibirapuera and Cerqueira César vary widely over time between 1 and 65 µg m$^{-3}$, approximately, but the two time series are highly correlated, with PM2.5 at Cerqueira César, a roadside site, exceeding that at Ibirapuera, a park site (though in a central area), by on average 4 µg m$^{-3}$. The US EPA 24-h PM2.5 standard of 35 µg m$^{-3}$ is exceeded on several occasions[53].

Panels b (filter measurements) and c (beta continuous) show PM2.5 mass concentrations in Ibirapuera, Pinheiros and the university campus again moving in step, driven by seasonal and meteorological shifts. Particle levels in Pinheiros, a roadside site, exceed those in Ibirapuera and the university campus. The IPEN-USP site on the university campus has little road traffic in its immediate vicinity.

Panel d compares beta-continuous PM2.5 measurements in Congonhas, a roadside site that is also near an inner city airport, against those in roadside Pinheiros. Judged by this beta-continuous fine-particle data, Pinheiros' air appears rather more polluted than that in Congonhas.

Panel e indicates that, at the same Pinheiros monitoring site and on the same dates, beta-continuous measurements of PM2.5 averaged over 24 h exceed 24-h filter measurements of the same particle range. This is consistent with the pattern in panel d suggesting that beta-continuous measures in Pinheiros appeared high relative to those in Congonhas. (We are not in a position to examine the reason for this divergence, but one possibility is calibration—the absorption coefficient—of the beta-instrument at the Pinheiros site, compared to the more reliable 24-h filter measurement.)

Finally, panels f and g show that BC mass concentrations measured during the field campaign at the university campus are highly correlated with PM2.5 mass concentrations measured by the environmental authority, both in the nearby IPEN-USP site in the same university campus (panel f), as well as in the nearby Pinheiros site across the Pinheiros river (panel g).

Such patterns reassure us as to the quality of the separate particle samples. Further description of the environmental authority's sampling sites, including aerial pictures, is available elsewhere[33, 48]. The submicron particle sampling site is located nearby to the environmental authority's IPEN-USP monitoring station.

**Gasoline versus ethanol mix in the active flex-fuel fleet.** Throughout the sample period, the composition of light-vehicle fuels that were ubiquitously dispensed across the metropolis' retailers, typically via a different nozzle at the same pump, were: ethanol E100, pure but hydrated, i.e., containing up to 4% of water; and gasoline E20 or E25, containing a 20 or 25% volumetric proportion of anhydrous ethanol. We refer throughout to E100 and E20/E25 by their consumer label, at retail, i.e., "ethanol" and "gasoline," respectively.

The gasoline blend, mandated nationwide by the federal government, was slightly modified on four occasions during our sample period. The blend shifted from E25 to E20 for purchases by retailers beginning February 1, 2010; from E20 back to E25 beginning May 1, 2010; again from E25 to E20 beginning October 1, 2011; and again from E20 back to E25 beginning May 1, 2013. The gasoline blend was consistently E20 during the DMPS sampling campaign that ran from October 2010 to September 2011, while it varied slightly during the PM2.5 sample period between November 2008 and May 2013. There were no other reported changes to the composition and quality of gasoline and ethanol fuels used by light-duty vehicles and motorcycles.

The vehicle fleet that was actively circulating in São Paulo city in July 2011 has been estimated at 5.9 million light vehicles (passenger vehicles including sport utility vehicles, minivans and light pickup trucks), 0.9 million motorcycles and 0.3 million heavy vehicles (trucks and buses)[33]. Whereas older light vehicles, sold prior to 2005, were predominantly equipped with single-fuel gasoline engines, the overwhelming majority of light vehicles sold after 2005 were "flex-fuel" gasoline–ethanol vehicles, transitioning between gasoline and ethanol combustion, according to consumer preferences, as relative fuel prices varied. Flex-fuel vehicles accounted for a likely but unknown share of total light-vehicle distance traveled within the São Paulo metropolitan area of "over (if not well over) 50% by 2011"[33]. Completing the fuel mix, motorcycles and heavy-duty vehicles were powered predominantly by gasoline and diesel, respectively, during the sample period. Variation in diesel prices and combustion over the sample period is discussed separately below.

Our empirical method is not based on trends in fuel consumption or trends in site-specific particle levels. We control for such potentially confounding trends in our regression models. The predominant combustion of gasoline among single-fuel light vehicles, and of diesel among heavy vehicles, are then interpreted as background levels of emissions that were unlikely to vary with the ethanol price fluctuations that occurred over the space of months.

We follow an earlier study[33] and, as the first step in a two-step model, construct the second step's main explanatory variable of interest, the gasoline share in the flex-fuel light-vehicle fleet from a consumer demand model, namely a multinomial probit choice model[30, 33]. We predict this time-varying market share of flex-fuel vehicles fueled with gasoline over ethanol based on gasoline and ethanol prices observed at the pump in São Paulo city during the sample period. To this end, we

obtained a large weekly panel of fuel prices, detailed by fuel pumping station and day the pumping station was surveyed, from the National Agency for Oil, Biofuels and Natural Gas (ANP; http://www.anp.gov.br/wwwanp/). This first-step demand model is estimated using actual consumer choices as a function of observed fuel prices and consumer demographics[30]. It is important to realize that the reason why we need a demand model to predict day-to-day fuel quantities from day-to-day fuel prices is that high-frequency fuel quantity or usage data for the metropolitan area of São Paulo are not available, only price data. Otherwise, we would skip the first-step model and use the fuel quantity data directly.

To account for fuel stored in vehicles' tanks, following consumer purchase but prior to combustion, we use four-day lagged prices at the pump. Previous research documents that the median consumer purchases fuel once a week[30]. Thus, the gasoline share of combustion on day $t$ is predicted from fuel prices at the pump $7/2 \approx 4$ days earlier. In a robustness test, we increase consumer stocks to 7 days.

The predicted gasoline share in the flex-fuel vehicle fleet, denoted by $\hat{s}_t^{\mathrm{gas}}$, ranges from a sample minimum of 0.14 in spring 2009, and similarly in spring 2010, to a sample maximum of 0.76 in late summer/early fall 2011 (Supplementary Fig. 4a). The hat in $\hat{s}_t^{\mathrm{gas}}$ indicates that the gasoline share is estimated from the first-step model. Correspondingly, the ethanol share (one minus the gasoline share) among flex-fuel vehicles fluctuated between 0.86 and 0.24 at these points in time.

Denoting the retail prices of 1 litre of ethanol and one litre of gasoline by $p_e$ and $p_g$, respectively, the evolution of the price ratio $p_e/p_g$ mirrors that of the predicted gasoline share, $\hat{s}_t^{\mathrm{gas}}$, over the 2008 to 2013 sample period (Supplementary Fig. 4b). The fact that the fuel mix $\hat{s}_t^{\mathrm{gas}}$ and the relative price $p_e/p_g$ move together reflects the previous finding that the relationship between the gasoline (or ethanol) choice probability and the ethanol-to-gasoline price ratio is quite linear over a wide range of price variation[30]. In particular, consumer preferences and behavior are such that flex-fuel vehicle drivers, who are overwhelmingly household consumers, do *not* as a whole transition abruptly between gasoline and ethanol at the relative price point at which the effective prices of ethanol and gasoline, in $/km of distance traveled, are equalized.

Instead, consumer switching is significantly more gradual—or demand is less elastic—around this parity price ratio or threshold, that lies just under 0.70 for most vehicle models. To further describe consumer substitution patterns, as the price of ethanol rises slightly from a very competitive level (e.g., $p_e/p_g = 0.58$, or 0.70/1.2), some flex-fuel vehicle consumers already transition out of ethanol into gasoline, despite ethanol still remaining very competitively priced. As the price of ethanol rises further and further, reaching a very uncompetitive level relative to gasoline (e.g., $p_e/p_g = 0.84$, or $0.70 \times 1.2$), some flex-fuel vehicle consumers still stay with ethanol at the pump.

The variation in the consumer price of ethanol relative to gasoline was observed throughout São Paulo. The resulting variation in the fuel mix that our work takes advantage of, with drivers induced to switch to gasoline and back to ethanol, was not isolated to specific neighborhoods. Any changes to particle emissions and secondary particle formation[54] that were a result of transitions between gasoline and ethanol combustion were happening at the citywide level (more precisely, at the state level), including the air surrounding each of the particle sampling sites. Moreover, ethanol price movements were the result of developments in world food and energy markets, rather than concerns over air pollution in São Paulo, which would otherwise make the main regressor of interest, the gasoline-ethanol mix, an endogenous variable (i.e., responding to the system we model, rather than exogenous to it).

Also following earlier work[33], our regression analysis drops the colder months of June to September. Seasonal variation in pollution tends to be pronounced[56, 57] and, importantly, the two episodes of marked ethanol price variation occurred outside these months (Supplementary Fig. 4b). Intuitively, we wish to keep the "high ethanol price, high gasoline share" days as otherwise comparable as possible to the "low ethanol price, low gasoline share" days. Including the colder months of June to September might introduce unobserved heterogeneity to this comparison.

As an alternative to the predicted gasoline share of consumer purchases at the pump, which is a series that varies daily based on daily prices for the city of São Paulo, we also compute a lower-frequency, more-regional gasoline share from aggregate quantity data, available from ANP (Supplementary Fig. 4c). Denote this gasoline share by $s_t^{\mathrm{gas,aggr}}$, where the absence of a hat indicates that the share is calculated, not predicted, from data. This alternative measure of the fuel mix is computed from monthly, and possibly incomplete, fuel shipments reported by wholesalers for the state (not city) of São Paulo. Wholesale quantities of blended gasoline (E20/E25) and hydrated ethanol (E100) are reported separately, in cubic meters/month. Prior to computing aggregate shares, we adjust for differences in energy content by converting the separate fuel quantities in cubic meters to light-vehicle distance traveled, given assumptions on the fleet's fuel economy.

The alternative aggregate wholesaler gasoline share, $s_t^{\mathrm{gas,aggr}}$, varies less than the baseline gasoline share, $\hat{s}_t^{\mathrm{gas}}$, since the latter relates to choices in the subpopulation of flex-fuel vehicles whereas the former includes gasoline and, to a lesser extent, ethanol consumption by single-fuel light vehicles and motorcycles. Importantly, $s_t^{\mathrm{gas,aggr}}$ moves in step with $\hat{s}_t^{\mathrm{gas}}$. For example, $s_t^{\mathrm{gas,aggr}}$ also reaches a sample minimum in spring 2009 and a sample maximum in late summer/early fall 2011. That $\hat{s}_t^{\mathrm{gas}}$ (a high-frequency series predicted from high-frequency price data) and $s_t^{\mathrm{gas,aggr}}$ (a low-frequency series based on data for the state)—move in tandem heightens our confidence in using the high-frequency $\hat{s}_t^{\mathrm{gas}}$ as the preferred specification for our main explanatory variable of interest.

Earlier work[33] described the "large-scale switching out of ethanol and into gasoline as ethanol prices soared, and back to ethanol when prices dropped," as indicated by the ANP reports between 2009 and 2011: "wholesaler reports suggest that the unblended (pure) gasoline component shifted between 42 and 68% of total gasoline-plus-ethanol light-vehicle distance travelled". This represented a 60% increase in the pure gasoline share (68/42-1), equivalent to a 45% reduction in the pure ethanol share (1–32/58). The additional evidence agrees that the change in the fuel mix was massive. To summarize, the alternative measure $\hat{s}_t^{gas,aggr}$, based on aggregate monthly wholesale reported quantities for the entire state's fleet, serves as a robustness check on the gasoline share in the flex-fuel vehicle fleet $\hat{s}_t^{gas}$ that is predicted from the high-frequency price series.

**Diesel combustion in the heavy vehicle fleet**. We argue that diesel combustion in heavy vehicles, while an important contributor to particle emissions and secondary particle formation, is unlikely to confound our inference of the effect on particles of gasoline vs. ethanol use in light vehicles during the periods we examine. We begin by considering variation in the retail price of diesel oil in the São Paulo metropolitan area between November 2008 and May 2013, available from the Brazilian Institute for Geography and Statistics (IBGE, Supplementary Fig. 5).

After a downward 5% price adjustment in mid 2009, diesel prices stayed constant in nominal (inflation-unadjusted) terms, and gradually declined in real (inflation-adjusted) terms, until mid 2012, when the federal government began to partially adjust diesel prices for cumulative inflation observed in the preceding years. In real terms, diesel prices in May 2013 were still below their October 2008 level, as was the case for gasoline prices. In particular, diesel prices hardly changed in nominal terms (and hardly changed beyond a gradual downward trend in real terms) during the DMPS sampling campaign that ran from October 2010 to September 2011.

In contrast to pronounced fluctuations in the price of ethanol, the subdued variation in the price of diesel—including the absence of fluctuations—suggests that diesel use is unlikely to be a confounder in our regression analysis. If anything, diesel prices in real terms followed a gradual downward trend over several years, and our particle regression models include a time trend, which absorbs the effect of any omitted determinant of particle concentrations that exhibits a trend. Controlling for diesel prices in the particle regression, as we do in robustness tests, indeed does not change our results.

We provide three additional pieces of evidence to underscore the point that omitted variable bias due to variation in diesel combustion is unlikely to be present. First, buses in the public transport system are a key source of diesel emissions in the São Paulo metropolitan area. From São Paulo's public transportation authorities (SPTrans), we obtained monthly ridership on buses in the public transport system across the metropolis from November 2008 to May 2013 (Supplementary Fig. 6). Ridership was quite stable over the period, tending to fall in the month of January in the yearend school vacation period, and similarly in the winter month of July in which schools also break (these days are either controlled for using separate type-of-day fixed effects, or excluded from our regression samples). There is no indication that commuting on (use of) diesel buses responded to the gradual decline in real diesel prices (Supplementary Fig. 5), consistent with the provision of public transport being insensitive to diesel prices (which hardly varied in the first place). Moreover, there is no indication that flex-fuel vehicle motorists might have taken to public transport as ethanol prices rose, which could otherwise confound our inference. Controlling for diesel bus ridership in the particle regression, as we do in robustness tests, does not change our results (Fig. 3 compared to Fig. 1).

Second, from SPTrans we further obtained the actual frequency of public transit diesel buses passing through the university campus where the submicron particle sampling site was located, during the sample period between October 2010 and May 2011 that we use to estimate our submicron particle regression models. The data come from billing records and are for realized trips in both directions. The number of diesel buses passing within a horizontal distance of 400 m from the site on a weekday morning (09:00 to 09:59) was stable at about 25 h⁻¹ from October 2010 to March 2011 (Supplementary Fig. 7). This is about one diesel bus every 2 min, underscoring the limited influence of local sources on the fourth-storey site. Moreover, bus line 8012–10 was added on March 29, 2011, increasing the number of diesel buses in April and May 2011. Our finding that ultrafine particle levels fell during these months of expanded bus service on campus further underscores the limited influence of local sources on the submicron particle site (and both series are uncorrelated during most of the sample period).

Indeed, in sensitivity analysis we include the (always low) observed diesel bus frequencies on campus as an additional control and show that the estimated effect of gasoline penetration on ultrafine particle levels is robust, and even strengthened (Supplementary Figs. 15 and 16). The intuition is that, contrary to diesel bus frequency, gasoline use varies along with ultrafine particle levels. Beyond the October 2010–May 2011 diesel bus frequencies observed from SPTrans, we obtained student enrollment at the USP Armando Salles de Oliveira campus between 2009 and 2013. This should inform on any variation in the demand for diesel bus services on the university campus. Enrollment over 2009–2013 has been very stable, for example, undergraduate enrollment varied by no more than 50 students around a mean enrollment of 7,451 students. Regular circulation of campus buses, or diesel vehicles anywhere, contribute to background levels of emissions that were unlikely to vary with the ethanol price fluctuations.

Third, we obtained monthly diesel fuel shipments reported by wholesalers for the state of São Paulo—the same ANP data source as the wholesale gasoline and ethanol fuel shipments described above[58]. Unfortunately, these diesel shipments include the large and seasonal statewide highway market; diesel volumes specific to the São Paulo metropolitan market are not publicly available. State-level diesel shipments over the course of the first semester of 2011 broadly followed their typical upward seasonal trend (Supplementary Fig. 8). Importantly, there is no evidence of confounding correlation with the pronounced fluctuation in the price of ethanol relative to gasoline and in the gasoline share—namely up until the beginning of April 2011, and down thereafter—that we exploit in our empirical analysis (noting, again, that our regression models control for trends).

In view of a recent literature that studies the effect on particle emissions of introducing biodiesel as a substitute for diesel[59–62], we note for completeness that the biodiesel fraction is low and changed only slightly in July 2009, from 3 to 4%, and in January 2010, from 4 to 5%. In particular, the diesel-biodiesel mix did not change during the submicron particle sampling period and is unlikely to confound our estimates. Moreover, the slight changes in diesel composition that happened earlier were mandated nationwide by the federal government, and were not policy responses to fluctuations in particle pollution in São Paulo.

In sum, the evidence indicates that potentially confounding effects on the particle size distribution from variation in heavy vehicle traffic around the time of each ethanol price hike are unlikely.

**Meteorological and atmospheric conditions**. Beyond the gasoline-ethanol fuel mix that is our focus, meteorology, including the occurrence of thermal inversions, is a key determinant of particle pollution. To control for possible confounders and increase estimation precision, we obtained hourly meteorological data recorded at weather stations run by the environmental authority (CETESB)[48] and by the Institute for Meteorology (INMET, http://www.inmet.gov.br), located in different parts of the metropolis. This follows earlier work on gaseous pollutants,[33] which describes the hourly meteorological data, provides an overview of meteorology in São Paulo including spatial correlation across the metropolis, and examines how meteorology is strongly associated with $O_3$, NO, CO, and PM10 concentrations. In addition to the weather controls used in earlier work, we control for the occurrence and height of thermal inversions in the lower atmosphere. These are recorded every 12 h, at 09:00 and 21:00 local time, by the Brazilian Air Force (FAB)[48].

**Local vehicle traffic conditions**. To control for vehicle traffic in the area surrounding a particle sampling site as well as across the city's road grid, we use hourly traffic congestion records from the city's traffic authority (CET; http://cetsp1.cetsp.com.br/monitransmapa/agora/). These are available at the road segment level (approximate length 100 m) for an 840-km grid of monitored roads and corridors across São Paulo city. As previous work points out, "(a) concern that might arise in a real-world—as opposed to lab or synthetic—setting such as ours is the possibility that consumers may have cut back on vehicle usage when faced with rising ethanol prices"[33]. Previous research finds that vehicle usage, as measured by road traffic congestion and speeds, did not fluctuate with the ethanol-to-gasoline price ratio that drove the fuel mix. Nevertheless, all the particle regression models we estimate control for possibly confounding variation in road congestion.

Specifically, the evidence suggests that commuters did not change their travel behavior as ethanol prices rose, such as drive less or switch from light vehicles to public transport. A previous study concluded that "(r)ising ethanol prices, beginning in mid 2009 and again in mid 2010, did not ease traffic congestion, raise traffic speeds, or increase ridership in the public transportation system. Similarly, when ethanol prices began falling in March 2010 and again in April 2011, motorists did not take to their vehicles more often"[33]. One can interpret this empirical finding on the basis of the poor availability of short-run substitutes to a commuter's adopted mode and distance of travel, the relatively stable price of gasoline fuel (a substitute for most consumers of ethanol fuel), and the state of "repressed demand" for road space in the face of widespread gridlock across the São Paulo metropolis.

This earlier study also described patterns in the road traffic data, including the daily and weekly commuting cycles, the annual calendar of public holidays, and the yearend school vacation fortnight that typically starts on December 24 and during which traffic might flow a bit more freely. To illustrate, aggregated across a citywide 840-km grid of monitored roads and corridors, records show that the total extent of congested road segments peaks at 09:00 during the workday morning commute (at 82 km of traffic extension), and again at 19:00 during the workday evening commute (121 km). Beyond workdays, congestion is relatively high (though at lower levels) at 14:00 on a non-holiday Saturday (16 km), at 19:00 on a public holiday (14 km) and at 20:00 on a non-holiday Sunday (4 km). These statistics are means across dates in the sample period 2008 to 2013.

We can also use the detailed traffic congestion data to rank the particle sampling sites in terms of their proximity to vehicular traffic. Averaged across all non-holiday weekdays in the 2008 to 2013 sample, road traffic congestion recorded at 09:00 within a 2 km radius of each site is, in increasing order: USP (0.5 km of traffic extension); Cerqueira César (1.8 km); Pinheiros (4.4 km); Ibirapuera (6.6 km); Congonhas (7.6 km). We note that while the Ibirapuera site is located in a park, busy (traffic-monitored) roads surround the park itself.

In addition to road traffic, we obtained the number of aircraft take-offs and landings hour by hour at Congonhas airport, obtained from the National Agency for

Civil Aviation (ANAC). In principle, controlling for such variation may be most relevant to explaining particle levels in the neighboring Congonhas air monitoring site.

**First-step inference in a two-step regression model**. We examine particle measurements through the lens of our two-step multivariate regression model[33]. In a first step, the gasoline share—the proportion of flex-fuel vehicles fueled with gasoline E20/E25 rather than ethanol E100—is constructed using daily gasoline and ethanol prices and a demand model that was estimated using actual consumer choices[30]. Intuitively, we used observational data on how consumers actually substituted gasoline for ethanol and back when prices fluctuated in 2010, to predict how they substituted gasoline for ethanol and back when prices fluctuated during the submicron particle sampling period in 2011 (Supplementary Fig. 4). Since the gasoline share is a prediction from this first-step model, the confidence intervals in our second-step particle regression estimates need to account for this sampling variation[40]: widening the confidence intervals on the estimated effects of shifting the gasoline vs. ethanol mix on ambient particle levels is in effect what the bootstrap procedure[63] described below achieves. As stated, the reason we need a demand model to predict day-to-day fuel quantities from day-to-day fuel prices is that the former are not available for the São Paulo metropolis; otherwise, we would use the fuel quantity data directly in the second step and skip the first-step model.

**Second-step particle regression model specifications**. In a second step, we investigate how mass concentrations of PM2.5 and of BC as well as PM number concentrations in the 7–800 nm size range change as the proportion of gasoline in the fuel mix varies, holding other factors constant. We fit empirical models of the form (Table 4):

$$\text{particles}_{lt} = \text{fuel\_mix}_t'\lambda + W_{lt}'\Delta^W + A_{lt}'\Delta^A + T_{lt}'\Delta^T + \upsilon_t + \mu_l + \varepsilon_{lt} \quad (1)$$

An observation is a measurement location $l$ by time period $t$ pair, or simply a time period $t$ for regressions estimated for a single location. The dependent variable particles is the field measurement—namely particle mass concentration or particle count according to our different data sets—that we seek to explain on the basis of temporal variation in the gasoline-ethanol fuel mix, fuel_mix, and other temporal and spatial determinants of particle pollution, namely meteorological, atmospheric, and road traffic conditions, respectively denoted by vectors $W$, $A$ and $T$. Intuitively, we seek to uncover the co-variation between (say) ultrafine particle levels in ambient air and the gasoline share, after correcting for differences in other determinants of particle levels, such as meteorological, atmospheric, and road traffic conditions, and fixing season, day of the week and time of the day. In addition to road traffic, we include controls for aircraft traffic in $T$ when examining PM2.5 measured near Congonhas airport (Supplementary Table 3).

Fixed effects $\mu_l$ and variables $\upsilon_t$, respectively, capture omitted location-varying and time-varying drivers of particles. These include indicator variables to account for cyclical effects at the annual level (seasonality, by week or quarter of year) and weekly level (within-week commuting and trade patterns, by type of day), as well as site-specific time trends or year fixed effects, to account for secular changes in economic activity or in road and fleet composition. For example, specifying a separate binary variable (intercept) for each day of the week, included in vector $\upsilon_t$, allows mean particle levels at a given time on Sundays, as explained by the model, to differ from mean particle levels on Mondays.

Of potential relevance to some locations and the longer samples, we include site-specific binary variables to indicate dates after the opening of the southern section of the Greater São Paulo beltway, on March 31, 2010. The beltway inauguration may have shifted the composition of road users, with less diesel-burning heavy vehicles circulating in the inner city after March 2010, as they could now use the beltway. Estimates for PM2.5 and ozone since 2008 in Tables 2 and 3 correct for this potential omitted variable. In sensitivity analysis, to vector $T$ we add diesel bus ridership or the real price of diesel in the São Paulo metropolis, or public transit bus frequency on the USP campus, and show that our estimates hardly change (Supplementary Figs. 8, 15 and 16). In effect, these proxies for diesel combustion did not vary around a trend or in step with the gasoline-ethanol mix.

Regression model (1) is estimated by ordinary least squares (OLS). $\lambda$, $\Delta^W$, $\Delta^A$ and $\Delta^T$ are coefficients to be estimated, and $\varepsilon_{lt}$ is an econometric residual. The identifying assumption is that, conditional on controls $X_{lt} := (W_{lt}, A_{lt}, T_{lt}, \upsilon_t, \mu_l)$, the residual is uncorrelated with the fuel mix, in particular:

$$E[\text{fuel\_mix}_t \varepsilon_{lt}|X_{lt}] = 0, \text{ where } X_{lt} := (W_{lt}, A_{lt}, T_{lt}, \upsilon_t, \mu_l) \quad (2)$$

For our main variable of interest, fuel_mix, we consider the gasoline share $\hat{s}_t^{\text{gas}}$ predicted by a consumer demand model (Supplementary Fig. 4a)[30]. To account for sampling variation in generating a prediction for $s_t^{\text{gas}}$—an estimated rather than observed variable—we bootstrap the original sample of consumers observed making choices at the pump[30, 33]. For every one of 200 bootstrap samples of consumers, $b = 1,...,200$, we obtain a different gasoline choice probability $s_t^{\text{gas},b}$, for each different combination of ethanol and gasoline prices (i.e., a different day) in the particle sample. We then use these 200 first-step bootstrap consumer samples to make inference from our second-step particle regressions. What this means in

practice is that we obtain a slightly different set of estimated coefficients $(\hat{\lambda}^b, \hat{\Delta}^{W,b}, \hat{\Delta}^{A,b}, \hat{\Delta}^{T,b})$ for each bootstrap sample $b$ (the hat denotes an estimated, rather than known, parameter); the bootstrap standard error is then the standard deviation of the coefficients estimated over the 200 bootstrap samples. It is because we estimate rather than observe this measure of the fuel mix that we need to correct for sampling error in $\hat{s}_t^{\text{gas}}$ when reporting standard errors on the estimated coefficients of model (1)[40].

To illustrate how identification of the causal effect of the fuel mix (gasoline share) on particle pollution works, via condition (2), we describe a hypothetical example where it would fail. Consider the January to May 2011 submicron particle size distribution sample (Fig. 2). We find that ultrafine particle levels and the gasoline share rose in tandem from January 20 to late March, and similarly the two variables jointly declined over April and May. This is seen once we correct for other potential influences on ultrafine particles such as random variation in wind speed (Fig. 2d). Now suppose that, hypothetically, ultrafine particle levels are unrelated to the gasoline share and, instead, there exists a time-varying driver of ultrafine particle levels that the researcher is unaware of, which similarly moves in one direction from January to March (say up), and then moves in the opposite direction from April to May (say down). In this case, the researcher would mistakenly interpret the tight co-movement between the gasoline share and ultrafine particles as a causal relationship, whereas all that is being estimated is a correlation between the unknown time-varying omitted variable (the true driver in the hypothetical example), the gasoline share (the interpreted driver in the hypothetical example), and ultrafine particle levels. Formally, the confounding omitted variable in the example, whose influence is not corrected for, would remain in the econometric residual, $\varepsilon_{lt}$. Since this omitted variable, even after conditioning on controls, is (positively) correlated with the gasoline share, which is included as a regressor in the estimating equation, the identifying assumption would fail: in this hypothetical example, $E[\text{fuel\_mix}_t \varepsilon_{lt}|X_{lt}] > 0$.

As a first alternative to correcting for standard errors on estimates of particle regression model (1) by way of a bootstrap procedure, we can estimate the model by 2SLS. Taking advantage of the finding that over the relevant range fuel shares are highly correlated (and approximately linear in) the ethanol-to-gasoline price ratio (Supplementary Fig. 4b)[30], we use this price ratio as an IV for the estimated regressor $\hat{s}_t^{\text{gas}}$. The identifying assumption for this variant is then:

$$E\left[(p_e/p_g)_t \varepsilon_{lt}|X_{lt}\right] = 0, \text{ where } p_e/p_g \text{ is the ethanol } - \text{to} - \text{gasoline price ratio} \quad (3)$$

Importantly, ethanol prices were responding to developments in the world sugar market; in particular, price shocks can credibly be taken as exogenous to air pollution in the São Paulo metropolis—thus $(p_e/p_g)_t$ is likely to be uncorrelated with unobserved determinants of particle levels $\varepsilon_{lt}$, and is a valid instrument for the imputed (estimated) gasoline share regressor $\hat{s}_t^{\text{gas}}$. This specification (robustness) test based on an IV estimator may alleviate any concern with regard to the possible presence of measurement error in the gasoline share, or potential confounding from unobserved determinants of particle pollution, captured in the residual, that may correlate with the gasoline share but not with its instrument (the observed price ratio).

As a second alternative to using the gasoline share imputed for the São Paulo metropolis from an estimated consumer demand model, we use the gasoline share calculated from available aggregate monthly fuel quantity data reported by wholesalers for the entire state's fleet, $s_t^{\text{gas,aggr}}$ (Supplementary Fig. 4c). We estimate particle regression model (1) by OLS, with the identifying assumption:

$$E\left[s_t^{\text{gas,aggr}} \varepsilon_{lt}|X_{lt}\right] = 0 \quad (4)$$

Again, alternative measure $s_t^{\text{gas,aggr}}$ serves as a robustness check on the flex-fleet gasoline share $\hat{s}_t^{\text{gas}}$ that is predicted from the high-frequency price series, and both variables move in step.

Table 1 summarizes the alternative outcome variables, particles, that we examine, and the key regressor of interest $\hat{s}_t^{\text{gas}}$ (the second row in the section labeled "Fuel mix variables"), along with the alternative share $s_t^{\text{gas,aggr}}$ (two rows below). The table also describes the different control variables—determinants of particles levels other than the gasoline–ethanol fuel mix—that the multivariate regression corrects for. Table 4 provides an overview of all the estimated regression model specifications, reported both in the main text and in the Supplementary Information. We list: the dependent variable of the regression equation (e.g., UFP 7–100 nm); the sample period and temporal aggregation of the data as employed in the regression (e.g., 1-h or 24-h mean); the main regressor of interest (e.g., $\hat{s}_t^{\text{gas}}$ or $s_t^{\text{gas,aggr}}$); the estimation procedure (e.g., OLS + bootstrap or 2SLS) and other sensitivity analysis provided.

**Data availability**. The data archive can be accessed at https://goo.gl/9tNzvj.

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

## Acknowledgements

We gratefully acknowledge numerous people from ANP, CET, CETESB, INMET, and SPTrans for generously sharing their data. In particular, we thank Rodrigo Sartoratto de Alencar, Wagner Baptista, Cristina Costa, Alaor Dall'Antonia Jr, Mauricio Lima Ferreira, Masayuki Kuromoto, Carlos Lacava, Dario Garcia Medeiros, Rui César Melo, Clarice Aico Muramoto, Roseni dos Santos, and Telma Paulino Senaubar. We thank Sam Ritchey and Jiaxiu He for listing the GPS coordinates of road segments. A.S. and F.M.G. acknowledge support from the Initiative for Sustainability and Energy at Northwestern University (ISEN) and the W Awards Program at Northwestern University's Weinberg College of Arts and Sciences. A.S. thanks the Dean's Office at the Kellogg School of Management, Northwestern University, and the Dean's Office at the Faculty of Arts & Social Sciences, National University of Singapore for support under start-up grant R-122-000-187-133. P.A. and J.B. acknowledge funding from FAPESP—Fundação de Amparo a Pesquisa do Estado de São Paulo through grants 2013/05014-0 and 2013/25058-1.

## Author contributions

All authors analyzed the data and wrote the paper.

## Additional information

**Competing interests:** The authors declare no competing financial interests.

