## [Peer Review File · Nature Communications]

Reviewers' Comments:

Reviewer #1 (Remarks to the Author)

This paper presents very important results based on the analysis of ambient air quality data to identify the actual effects of increased use of ethanol fuels in light duty vehicles on particulate matter, including ultrafine particulate in the air.

The manuscript (and supplementary materials) presents a very clear and detailed description of the approach to quantifying the changes in the ethanol portion of the fuel used in Sao Paulo as the price of ethanol changed due to market conditions. It also presents a very detailed description of the many factors that can affect ambient particulate concentrations. The analysis of this PM data shows that there is a statistically significant increase in the atmospheric particle count for ultrafine particulate during periods of increased gasoline use (decreased ethanol use). Other PM categories including PM100-800 nm, PM2.5 and BC showed no statistically significant change during periods of increased gasoline use. This study also shows that ozone concentrations decrease statistically significantly during periods of increased gasoline use (decreased ethanol use). This analysis produces similar results for the effects on ozone concentrations to those reported earlier by these authors (Salvo and Geiger, *Nature Geoscience*, 7,450-458 (2014)).

The approach used in the analysis of this data is novel. The quality of the data used in the study are demonstrated by the correlation plots presented in the supplementary materials. The methodology used is presented clearly and with considerable detail in the manuscript and supplementary materials. The effects of meteorological variables on the concentrations of PM and ozone found in this analysis are consistent with the general expectations of these effects. The analysis techniques used here seem to deal well with the many factors that affect ambient concentrations, and allow the authors to extract information on the effects of differences in the fuels used on the ambient concentrations observed.

The statistical and analytical techniques used in the analysis of the ambient data are quite appropriate and robust and are expected to provide valid and reliable results. The manuscript presents a clear and appropriate description of a very complex analysis. I have no suggestions for improvements. This manuscript is suitable for publication without modification.

Reviewer #2 (Remarks to the Author)

Comments to the Author

General comments:

This paper presents interesting research on an important topic of transport environmental management and modeling techniques in a Brazilian case study with black carbon BC, O₃ and particle matter PM_{2.5}/UFP vehicle related emissions using an econometric model. The model tried to demonstrate the influence of the fuel shift, from ethanol to gasoline, on particle emissions - especially nano-particles during morning peak hours. The paper is well written and presents the results in a clear and concise way. However, some of the main conclusions are not well supported (by the way, why the text has no numbered lines?). It seems there is not a lot of new methodology compared to former authors' studies. The authors are using the econometric model applied to the Brazilian/São Paulo air quality database (CETESB, IF/USP, INPE, etc.) and it appears that other modeling approaches could be more suitable. At least, the authors should mention other available approaches and/or models (especially regarding real world emission models and source identification techniques).

This could be a useful paper for transport managers and policy makers, but to tempt people to understand it, there is a major need to make it more focused on the work of the authors (i.e. in tables 1, 2 & 3 the result estimates coming from the study are not adequately presented and/or

are difficult to follow). You could also compare your results with some national/international studies on emission factors and source analyses. There are also some conceptual issues that must be clarified (i.e. Heavy Duty Vehicles are neglected from Ultrafine Particle UFP Emissions). Try to justify some sections and explain better methodology, temporal scenario-settings and data (included a summary table with all sample methods, timing, species, etc.).

Regarding the modeling part (econometric model), although it is easy to understand the model estimates, if you review the concluding tables 1 to 3 and the supplementary material, and the calculations made which leads to the author's conclusions, it is needed to perform a deep analysis on the key model parameters. The authors should better justify some of the assumptions used (i.e. no contribution of diesel fuel consumption on UFP changes). I don't think all the remarks within the discussion section are either necessary or the arguments given are vague. Better complement or remove unnecessary paragraphs.

You should compare your results with some other studies or tests; you should also compare the modeling results with additional observation measurements, at least with some empirical data from the literature and/or former studies in the Metropolitan Region of São Paulo. The contents of the paper are mixed up and it is difficult to know what is coming directly from the modeling results and what are general author's outcomes and a priori hypothesis (see the section discussion).

Why using the econometric model for solving temporal/spatial scenarios and not other model/algorithms? Please justify your choice. The author may also need to consider whether additional tables are needed in the context of environmental and public research journal focused on environmental and vehicle emission issues. The structure and text of the paper is easy to understand. The three tables included in the text have too much information. In my opinion, it is needed a major review.

Specified comments:

Abstract:

- With higher ethanol prizes, nano-particles increase by 30%. What about diesel and/or biodiesel vehicle consumption changes, in absolute and relative terms?
- Short researched period of significant fuel-fleet shift - 11 months - and only one spot (IF/USP).
- Concerning urban ozone, must be another reasons to this increase as biofuel consumption increment, which could increase NOx emissions and formation of secondary particles (as nano-particles).

Chapter Main text:

- The influence of Heavy Duty Vehicles (HDVs) could not be neglected (outnumbering heavy-duty diesel vehicles by an order of magnitude or so), looking at the fuel sale statistics in São Paulo. Also looking at the spot where you have nano-particle data, with is strongly influenced by traffic of buses especially during peak hours (see Figures s8 and s9 of the supplementary material).
- Although gasoline combustion can lead to larger amounts of ultrafine particles UFP (<100 nm in diameter) when compared to ethanol combustion, there are other studies, which demonstrate the influence of ethanol on secondary aerosol formation.^{1,10}
 - Biodiesel and diesel vehicular emissions have also influence on secondary ultrafine particle formation.^{2,3,4,5} There was need of improvements in after treatment technologies to reduce pollutants emissions, as could be the case of HDVs at the USP Campus in 2011 (maybe now in 2016 the vehicles are newer and the situation have changed, even after increasing the use of biodiesel in the fuel blends). Please review these and other articles.
 - There are also other studies, which compare diesel vs. gasoline vehicle engines;⁶ research UFP emissions from fuel characteristics and after treatment technologies;⁷ research UFP emissions from non-fuel vehicle components (brakes and tires);⁸ and research the influence of traffic conditions on ultra-fine particle emissions.^{9,10}

- It seems there are many confounding effects on top of temperature, wind speed, boundary layer height, the spatial distribution of traffic, and even drifts. So, why using the econometric approach among others? You could use a different methodology, trying to identify sources and more focus on trace elements. Source distribution analysis of PM_{2.5}/BC/element measurements from the researched sites with 24h filters could be performed using the Positive Matrix Factorization PMF of the Environmental Pollution Agency EPA for instance. At least comment other methodologies and possibilities.

- Do they import sugarcane in São Paulo from India? I think this remark is not so important since harvest conditions in India likely do not affect Brazilian ethanol production neither international imports.

- Again, it is not complete that area attribute particle sizes below 100 nm are only related to direct emissions from light duty vehicles LDVs; other studies relate the emission of nano-particles and bio-diesel and diesel consumption from trucking.

- You mention that shifts from gasoline to ethanol use increased ozone concentrations. Increase of O₃ concentrations could also be related to NO_x emission changes from HDVs and subsequent diesel consumption.¹¹

- You mentioned a plausible scenario where ultrafines fell while PM_{2.5} remained invariant. Could also be possible another scenario: global decrease of PM_{2.5} concentrations simultaneous to nano-particles?

Chapter Ultrafines rose with a shift to gasoline, and fell as consumers returned to ethanol:

- Ultrafine variation was observed during mid-summer to mid-fall (January 20 to May 31) of 2011 (5 months). The period was shorter than the 11 month observations included in the previous section (5 months, Oct. 2010-sept 2011). I found confusing the timing of the different samplings, sites and species (be consistent throughout the manuscript). I would recommend including a summary table with all these data.

- What does it mean a placebo test. Please omit subjective terminology. What about diesel/bio-diesel consumption rates?

- In São Paulo there are studies observing a different fact on a long-term: overall increase of O₃ levels and decrease of NO_x concentrations, despite the increment of gasoline fuel use.¹¹ In a former study you explained the same evidence: towards O₃ reduction in the afternoon by the fuel shift from ethanol to diesel.¹² What about other side effects as NO_x increments from HDVs?

Chapter Tight co-variation during morning rush hour:

- Ultrafine concentrations due to increased gasoline in the early morning. During early morning weather conditions are more stable due to lower planet boundary layer (PBL) and lower wind speeds. Weather dispersion conditions could have worsened and/or bio-diesel consumption could have changed during the researched five months, masking your results. If fuel shift would be the main driving reason, I would also expect evening significant UFP increases during peak hours where weather conditions are unstable and may have not change much during these months. Do you have any explanation for that? In other studies it was found a different behavior with heavy-duty vehicle (HDV) traffic decreases: lower pollutant concentrations during the Evening Peak compared to the Morning Peak.

Chapter Variation in 24-hours means:

- Estimates from model with the ethanol-to-gasoline price ratio serving variable for the predicted gasoline share. Why not using fuel sale statistics on a daily basis instead of this proxy based on fuel prizes? What about obtaining fuel sales from fiscal data? Why using ethanol/gasoline relative prizes on a multinomial probit choice model? Sometimes users are reluctant to change fuel (inelastic demand instead of less elastic demand). How do you establish background levels of emissions that were unlikely to vary with the ethanol price fluctuations that occurred over the space of months? The explanation given in the text is confusing.

Chapter Discussion:

- Absence of a significant relationship outside this time window and size range. What about

changing driving conditions during the observed months (maybe you have more congestion during morning peak hours).

- Laboratory experiments cited earlier. Which ones?
- Insignificant association between the light-vehicle fuel mix and BC levels. Maybe the sources are different?

References

1. Atmospheric Environment 117 (2015) 200-211
2. Fuel Processing Technology 96 (2012) 237-249
3. Science of the Total Environment 409 (2011) 738-747
4. Fuel 134 (2014) 201-208
5. Science of the Total Environment 500-501 (2014) 64-71
6. Front. Environ. Sci. 3:82. doi: 10.3389/fenvs.2015.00082
7. Environ Sci Technol. 2014;48(3):2043-50. doi: 10.1021/es405687m.
8. Environ. Sci. Technol. 2013, 47, 8091-8092. dx.doi.org/10.1021/es401805r
9. Atmos. Chem. Phys., 16, 8559-8570, 2016. doi:10.5194/acp-16-8559-2016
10. Atmospheric Environment 107 (2015) 374e390
11. Journal of Geophysical Research: Atmospheres 120 (12), 6290-6304
12. Nature Geoscience 7, 450-458,doi:10.1038/ngeo2144 (2014).

Reviewer #3 (Remarks to the Author)

This is an interesting article attempting to link the shift of gasoline use to ethanol with the decreased ultrafine particles. There are many studies on the emissions of ultrafine particles from gasoline and ethanol combustion. This study assess the concentrations to reach to conclusion that that this shift show a decrease in ambient concentrations. There are a number of issues and the findings can be challenged. For example, the assessment is made based on the measurements on a limited location which can not be taken as representative of the complex cities like Sao Paulo. Moreover, ultrafine particles are short-lived particles meaning that the sites away from the sources will not be representing the real emissions from the road traffic as they will be subjected to transformation processes such as nucleation, coagulation and condensation. Moreover, the biggest challenge is that a difference of 8000 particles per cm⁻³ (unit value seems wrong on fig 1 - it should be perhaps 10³ (not 10⁻³) is taken to conclude the effect of this shift. The urban concentrations has much more variability in the concentrations (of the order of 10⁴ cm⁻³, meaning that this difference cannot be taken as a direct effect of shift in fuel and many factors could have contributed to this change (e.g. distance of the measurement from the road, size range considered; instruments (different instruments such as CPC and DMPS are used without harmonising their data as these instruments could provide different results while measuring the same level of concentrations) and the dispersion conditions led by the atmospheric stability conditions. A number of directly relevant papers are overlooked which could have helped to build the arguments (e.g., Ultrafine particles in Cities, Environmental International) and enrich the discussions. While I see this an interesting study, the article is not fit for journals like Nature due to a number of weak points.

Reviewers' comments (in italics):

Reviewer #1 (Remarks to the Author):

This paper presents very important results based on the analysis of ambient air quality data to identify the actual effects of increased use of ethanol fuels in light duty vehicles on particulate matter, including ultrafine particulate in the air.

The manuscript (and supplementary materials) presents a very clear and detailed description of the approach to quantifying the changes in the ethanol portion of the fuel used in Sao Paulo as the price of ethanol changed due to market conditions. It also presents a very detailed description of the many factors that can affect ambient particulate concentrations. The analysis of this PM data shows that there is a statistically significant increase in the atmospheric particle count for ultrafine particulate during periods of increased gasoline use (decreased ethanol use). Other PM categories including PM100-800 nm, PM2.5 and BC showed no statistically significant change during periods of increased gasoline use. This study also shows that ozone concentrations decrease statistically significantly during periods of increased gasoline use (decreased ethanol use). This analysis produces similar results for the effects on ozone concentrations to those reported earlier by these authors (Salvo and Geiger, Nature Geoscience, 7,450-458 (2014)).

The approach used in the analysis of this data is novel. The quality of the data used in the study are demonstrated by the correlation plots presented in the supplementary materials. The methodology used is presented clearly and with considerable detail in the manuscript and supplementary materials. The effects of meteorological variables on the concentrations of PM and ozone found in this analysis are consistent with the general expectations of these effects. The analysis techniques used here seem to deal well with the many factors that affect ambient concentrations, and allow the authors to extract information on the effects of differences in the fuels used on the ambient concentrations observed.

The statistical and analytical techniques used in the analysis of the ambient data are quite appropriate and robust and are expected to provide valid and reliable results. The manuscript presents a clear and appropriate description of a very complex analysis. I have no suggestions for improvements. This manuscript is suitable for publication without modification.

We thank Reviewer #1 for these encouraging comments, which we very much appreciate.

Reviewer #2 (Remarks to the Author):

General comments:

This paper presents interesting research on an important topic of transport environmental management and modeling techniques in a Brazilian case study with black carbon BC, O₃ and particle matter PM_{2.5}/UFP vehicle related emissions using an econometric model. The model tried to demonstrate the influence of the fuel shift, from ethanol to gasoline, on particle emissions - especially nano-particles during morning peak hours. The paper is well written and presents the results in a clear and concise way. However, some of the main conclusions are not well supported (by the way, why the text has no numbered lines?).

We thank Reviewer #2 for these introductory comments. At the Reviewer's request, we have added numbered lines.

It seems there is not a lot of new methodology compared to former authors' studies.

Reviewer #2 is correct to point out that, in an earlier study (Salvo and Geiger, Nature Geoscience, 2014), two of the authors used a similar econometric approach to examine how variation in the light-vehicle fuel mix affected ambient ozone concentrations. We acknowledge this on page 5 (and acknowledged this in the original version).

We view the previous use of the method to examine gaseous pollutant levels available from the official monitoring network as a strength. We respectfully add that this particular strength was pointed out by Sasha Madronich, who in his piece accompanying Salvo and Geiger (2014) referred to the "purely empirical approach" as "the gold standard for the type of analysis needed to evaluate the reliability of atmospheric chemistry models designed to simulate the effects of the transportation sector on air quality."

The authors are using the econometric model applied to the Brazilian/São Paulo air quality database (CETESB, IF/USP, INPE, etc.) and it appears that other modeling approaches could be more suitable. At least, the authors should mention other available approaches and/or models (especially regarding real world emission models and source identification techniques).

Given the Reviewer's comment, we now state on pages 4-5 that the empirical econometric approach we follow to examine field-derived particle size distribution measurements provides an important empirical benchmark for alternative approaches based on modeling and source identification. We cite Madronich (2014), who provides a discussion of how to view the econometric approach in that context.

Specifically, the 2016 Faraday Discussion on Urban Air Pollution, which two of the authors of the current manuscript attended, outlines a number of modeling approaches (along with their limitations and strengths) that are based on emissions inventories. Specifically, in Faraday Discuss., 2016, 189, 455-72, issues are presented that arise from using emissions inventories for predicting NO_x concentrations (Spatially resolved flux measurements of NO_x from London suggest significantly higher emissions than predicted by inventories, Adam R.

Vaughan, James D. Lee, Pawel K. Misztal, Stefan Metzger, Marvin D. Shaw, Alastair C. Lewis, Ruth M. Purvis, David C. Carslaw, Allen H. Goldstein, C. Nicholas Hewitt, Brian Davison, Sean D. Beevers and Thomas G. Karl *Faraday Discuss.*, 2016, **189**, 455-472)

In the context of addressing actual versus hypothetical fuel switches, we additionally cite the published issue (Faraday Discussion on Chemistry in the Urban Atmosphere, 2016, 189, 1-680):

“Our “purely empirical” (Madronich, 2014) approach provides a concrete benchmark for alternative approaches used to evaluate urban air pollution, specifically those based on emissions inventories, the analysis of exhaust emissions or smog chambers, source apportionment studies and chemical modeling” (Faraday Discussion 2016, Keogh and Sonntag 2011).

This could be a useful paper for transport managers and policy makers, but to tempt people to understand it, there is a major need to make it more focused on the work of the authors (i.e. in tables 1, 2 & 3 the result estimates coming from the study are not adequately presented and/or are difficult to follow). You could also compare your results with some national/international studies on emission factors and source analyses. There are also some conceptual issues that must be clarified (i.e. Heavy Duty Vehicles are neglected from Ultrafine Particle UFP Emissions). Try to justify some sections and explain better methodology, temporal scenario-settings and data (included a summary table with all sample methods, timing, species, etc.).

We thank Reviewer #2 for these comments aimed at improving the reader’s understanding. In response to these and comments below:

- (i) We have added a first table, under new Table I, which describes the **different datasets that our study combines**. The table reports the data source, the sample period and the frequency of measurement, and provides summary statistics for each variable, namely the number of observations, mean, standard deviation, minimum and maximum in the sample.
- (ii) We have added a second table, under new Table IV, that provides an overview of **all the estimated regression model specifications**. For each table and figure that reports regression estimates in the manuscript and its Supplementary Information, we list: the dependent variable of the regression equation (e.g., UFP 7-100 nm); the sample period and temporal aggregation of the data as employed in the regression (e.g., 1-hour or 24-hour mean); the main regressor of interest (the gasoline share in the flex fleet or in the aggregate fleet); the estimation procedure (e.g., ordinary least squares) and other sensitivity analysis provided.
- (iii) The estimates reported in the original Table II are now presented in Table SVIII, in the Supplementary Information section, and in an enhanced Figure 3 (now showing estimated hour-by-hour changes in both UFP 7-100 nm and PM 100-800 nm).
- (iv) By replacing a table of estimates (original Table II) – estimates that can still be seen by way of a figure (revised Figure 3) – with two summary/overview tables (new

- Tables I and IV), our hope is to make the paper more accessible. Again, we thank the Reviewer for pushing us in this direction.
- (v) In the same vein, we have carefully reread the main text and the Supplementary Information, including table notes and all sections that describe regression estimates. We have edited, added or removed statements with the aim of improving readability and understanding. The results section of the Supplementary Information, Part H, now begins with two paragraphs that start as follows: “We begin with some general comments on how to read the tables of results, Tables SI to SVIII, which are discussed in what follows.” We have added more intuitive remarks on the method and on the results that we obtain. For Reviewer #2’s convenience, many of these changes are temporarily marked up in yellow background.

With regard to Reviewer #2’s important comments above and below on diesel combustion as a determinant of ambient particles and thus on its potential to confound the analysis:

- (vi) We clarify that the use of our method does not rely on the contribution of diesel combustion in heavy-duty vehicles to UFP levels being small. We are fortunate to be able to take advantage of the fact that diesel prices and combustion – as important a source as it may be – did not fluctuate (co-vary) during the episodes of marked variation in ethanol-to-gasoline prices and gasoline-ethanol combustion.
- (vii) Nevertheless, motivated by Reviewer #2’s comments, we have obtained data on monthly diesel bus ridership in the Sao Paulo metropolis’ public transport system (in millions of passengers per day). A new figure in the Supplementary Information, S8, provides sensitivity analysis when we re-estimate all the regression models reported in Table I in the main text, now including, as an additional control, monthly diesel bus ridership. Our previous estimates are very robust, as we expect from point (vi).
- (viii) We view this result as reassuring and thank the reviewer for strengthening our manuscript. We have chosen not to change the estimates reported in the original tables and figures, to reflect this additional control, but would be happy to do so if requested by the Editor and/or Reviewers. For convenience, the original Fig. 1 in the main text and Fig. S8 are reproduced at the end of our response to Reviewer #2.
- (ix) The Supplementary Information also notes that our regression estimates are robust to alternatively controlling for the price of diesel, which hardly varied around a gradual trend (if any). Our regression models already included a trend, that is, we are not identifying effects from variable trends, rather from co-movement (up and down) among particle levels and gasoline versus ethanol penetration once we correct for other observed factors (e.g., meteorology) and trends.
- (x) Importantly, we have added a new 2-page Part D in the Supplementary Information, “Diesel Combustion in the Heavy-Duty Vehicle Fleet,” that carefully shows evidence pertaining to diesel combustion, and lays out the argument. The section includes three figures: (new Fig. S5) Evolution of diesel prices in the metropolis; (new Fig. S6)

Ridership on diesel buses in the public transport system in the metropolis; (new Fig. S7) Monthly wholesale diesel fuel shipments for the state of São Paulo. For convenience, we reproduce the new Supplementary Information Part D and accompanying figures at the end of our response to Reviewer #2.

- (xi) To the first paragraph of the Discussion, we now add: “Fifth, controlling for monthly ridership of diesel buses in the metropolis does not change our estimates, due to bus ridership not varying over the sample period (Supplementary Part D).” We add a similar statement on pages 6-7.

Regarding the modeling part (econometric model), although it is easy to understand the model estimates, if you review the concluding tables 1 to 3 and the supplementary material, and the calculations made which leads to the author’s conclusions, it is needed to perform a deep analysis on the key model parameters. The authors should better justify some of the assumptions used (i.e. no contribution of diesel fuel consumption on UFP changes). I don't think all the remarks within the discussion section are either necessary or the arguments given are vague. Better complement or remove unnecessary paragraphs.

Please see the preceding comments. All tables in the Supplementary Information now report the effect of a shift in the fuel mix scaled for a 30% to 80% (50 percentage point) increase in the gasoline share in the flex-fuel fleet.

You should compare your results with some other studies or tests; you should also compare the modeling results with additional observation measurements, at least with some empirical data from the literature and/or former studies in the Metropolitan Region of São Paulo. The contents of the paper are mixed up and it is difficult to know what is coming directly from the modeling results and what are general author’s outcomes and a priori hypothesis (see the section discussion).

Please see the preceding comments.

Why using the econometric model for solving temporal/spatial scenarios and not other model/algorithms? Please justify your choice. The author may also need to consider whether additional tables are needed in the context of environmental and public research journal focused on environmental and vehicle emission issues. The structure and text of the paper is easy to understand. The three tables included in the text have too much information. In my opinion, it is needed a major review.

Please see our response to the previous sections of comments, including:

- (i) Interpretation of coefficients in the estimated regression models (“*calculations made... perform a deep analysis on the key model parameters*”);
- (ii) Comparison of results to the studies cited by the reviewer; assumptions or controls (“*contribution of diesel*”);
- (iii) The use of an empirical econometric model;

- (iv) The readability of original Tables I to III.

Specified comments:

Abstract:

- *With higher ethanol prizes, nano-particles increase by 30%. What about diesel and/or biodiesel vehicle consumption changes, in absolute and relative terms?*

Please see the new Supporting Information Part D. In particular, Fig. S6 reports monthly ridership on diesel buses in the metropolis' public transport system, and Fig. S7 reports monthly wholesale diesel (including a small proportion of biodiesel) shipments for the state of São Paulo. With specific regard to biodiesel, the section states:

“For completeness, we note the following changes in the composition of diesel during the full sample period. These changes, mandated nationwide by the federal government, slightly increased the proportion of biodiesel added to diesel oil: (i) from 3% to 4% beginning July 2009; (ii) and from 4% to 5% beginning January 2010.”

We now also state on page 17 (Methods), citing all four references listed by the Reviewer: “In view of a recent literature that studies the effect on particle emissions of introducing biodiesel as a substitute for diesel⁵⁵⁻⁵⁸, we note that the biodiesel fraction is low and changed only slightly in July 2009, from 3% to 4%, and in January 2010, from 4% to 5%. In particular, the diesel-biodiesel mix did not change during the submicron particle sampling period and is unlikely to confound our estimates.”

- *Short researched period of significant fuel-fleet shift - 11 months - and only one spot (IF/USP).*

We do not view the “short researched period of significant fuel-fleet shift” of 11 months as a weakness, as one must be careful when comparing distant periods that may differ for unobserved reasons in ways that the analyst is unable to control for. In fact, with this concern in mind, we use an even shorter (“more seasonally homogeneous”) subsample, from January 20 to May 31, 2011, to generate the submicron results shown in Figs. 1 to 3 in the main text.

It is also important to note that “quasi-experimental” variation of this scale, in which two million actively circulating passenger cars transitioned from ethanol to gasoline – and then returned to ethanol – within the space of a few months, is rare. The fact that the price of ethanol relative to gasoline has hardly varied since 2011, all the way to the present day, 2016, illustrates this. (Figs. S4 and S5 show this lack of price variation between 2011 and 2013.)

Reviewer #3 also states that additional sampling sites would be beneficial. Additional instruments – multiple units – were not available for use by the authors. In the case of a real-world episode at the scale of a megacity, more instruments cannot be deployed retroactively. Kumar et al. (2014), the review cited by Reviewer #3, characterize the instrumentation and measurement technology for ultrafine particles by their “high cost for field deployment in sufficient numbers” as well as “lack of robustness for long-term unattended operation” (p.7). The rooftop of a four-storey building inside the USP Armando Salles de Oliveira campus was chosen

due to the well mixed urban air, with influence from representative road emissions sources but not excessive proximity to idiosyncratic emissions. Two of the co-authors have used the site extensively in previous research. For example, Backman et al. chose this same site, describing it as follows on p.11735:

“The measurements were made at the Armando Salles de Oliveira campus area of USP (Fig. 1). The campus area is a vast park, totalling an area of 7.4 km², **making the site ideal for tracking ambient aerosols, without strong local sources**. At the campus area buildings are scarce. Thus, air masses arriving at the station should be well mixed and make the measurements **representative of the ambient pollution burden of the city.**”

(On the diurnal cycle of urban aerosols, black carbon and the occurrence of new particle formation events in springtime São Paulo, Brazil, J. Backman, L. V. Rizzo, J. Hakala, T. Nieminen, H. E. Manninen, F. Morais, P. P. Aalto, E. Siivola, S. Carbone, R. Hillamo, P. Artaxo, A. Virkkula, T. Petaja¹, and M. Kulmala, *Atmos. Chem. Phys.*, 12, 11733–11751, 2012)

- *Concerning urban ozone, must be another reasons to this increase as biofuel consumption increment, which could increase NOx emissions and formation of secondary particles (as nano-particles).*

Elucidating the mechanism will require extensive, long-term field studies as well as detailed laboratory models, which is beyond the scope of our current work (recall our statement that “the chemical analysis of the nanoparticles ... did not occur during the period of the fuel switch we studied, would be an important next step ...”). Instead, we report on an empirical observation in the field which, in principle, includes all physics, chemistry, and biology of the system. Importantly, our hope is that by providing observational evidence, our contribution will be to stimulate subsequent research, by pointing out the next areas to investigate.

Chapter Main text:

- *The influence of Heavy Duty Vehicles (HDVs) could not be neglected (outnumbering heavy-duty diesel vehicles by an order of magnitude or so), looking at the fuel sale statistics in São Paulo. Also looking at the spot where you have nano-particle data, with is strongly influenced by traffic of buses especially during peak hours (see Figures s8 and s9 of the supplementary material).*

Please see our earlier response to the concern raised regarding the role of diesel combustion as a potential confounder. We do not neglect the influence of HDV. Diesel price and quantity data (new Supplementary Information Part D) indicate that HDV emissions did not fluctuate (co-vary up and down) with observed gasoline-ethanol prices and quantities. Specifying monthly ridership on diesel buses in the metropolis (SPTrans) as additional controls in our regression equations hardly affects our results, as we show thanks to Reviewer #2 (new Fig. S8).

We note that, to the best of our knowledge, the only “fuel sale statistics for São Paulo” that are publicly available are monthly wholesale diesel shipments for the state of São Paulo, including the state highway market. Diesel quantity data specific to the São Paulo metropolis

(and at higher frequency than monthly) are not available. We report and discuss this data in the new Supplementary Information Part D, including on page S11 “Fig. S7 shows that state-level diesel shipments over the course of the first semester of 2011 broadly followed their typical upward seasonal trend. Importantly, there is no evidence of confounding correlation with the pronounced fluctuation in the price of ethanol relative to gasoline and in the gasoline share – namely up until the beginning of April 2011, and down thereafter – that we exploit in our empirical analysis (noting, again, that our regression models control for trends).”

- *Although gasoline combustion can lead to larger amounts of ultrafine particles UFP (<100 nm in diameter) when compared to ethanol combustion, there are other studies, which demonstrate the influence of ethanol on secondary aerosol formation.1,10*

- *Biodiesel and diesel vehicular emissions have also influence on secondary ultrafine particle formation.2,3,4,5 There was need of improvements in after treatment technologies to reduce pollutants emissions, as could be the case of HDVs at the USP Campus in 2011 (maybe now in 2016 the vehicles are newer and the situation have changed, even after increasing the use of biodiesel in the fuel blends). Please review these and other articles.*

- *There are also other studies, which compare diesel vs. gasoline vehicle engines;6 research UFP emissions from fuel characteristics and after treatment technologies;7 research UFP emissions from non-fuel vehicle components (brakes and tires);8 and research the influence of traffic conditions on ultra-fine particle emissions.9,10*

We have now reviewed and cite the following studies recommended by Reviewer #2:

- In the context of “...particle number concentrations that coincide with the fuel shifts may be different if they were to be measured near road traffic or at the vehicle exhaust^{50,51}” (page 14):
 - Goel, A. & Kumar, P. Characterisation of nanoparticle emissions and exposure at traffic intersections through fast–response mobile and sequential measurements. *Atmospheric Environment* 107, 374-390, doi:<http://dx.doi.org/10.1016/j.atmosenv.2015.02.002> (2015).
 - Giechaskiel, B. et al. Vehicle Emission Factors of Solid Nanoparticles in the Laboratory and on the Road Using Portable Emission Measurement Systems (PEMS). *Frontiers in Environmental Science* 3, doi:10.3389/fenvs.2015.00082 (2015).
- In the context of “...a recent literature that studies the effect on particle emissions of introducing biodiesel as a substitute for diesel⁵⁵⁻⁵⁸” (page 17):
 - Karavalakis, G., Boutsika, V., Stournas, S. & Bakeas, E. Biodiesel emissions profile in modern diesel vehicles. Part 2: Effect of biodiesel origin on carbonyl, PAH, nitro-PAH and oxy-PAH emissions. *The Science of the total environment* 409, 738-747, doi:10.1016/j.scitotenv.2010.11.010 (2011).

- Hoekman, S. K. & Robbins, C. Review of the effects of biodiesel on NO_x emissions. *Fuel Processing Technology* 96, 237-249, doi:<http://dx.doi.org/10.1016/j.fuproc.2011.12.036> (2012).
- Rahman, M. M. et al. Particle emissions from biodiesels with different physical properties and chemical composition. *Fuel* 134, 201-208, doi:10.1016/j.fuel.2014.05.053 (2014).
- Tadano, Y. S. et al. Gaseous emissions from a heavy-duty engine equipped with SCR aftertreatment system and fuelled with diesel and biodiesel: assessment of pollutant dispersion and health risk. *The Science of the total environment* 500-501, 64-71, doi:10.1016/j.scitotenv.2014.08.100 (2014).
- In the context of “Any changes to particle emissions and secondary particle formation^{60,61} that were a result of transitions between gasoline and ethanol combustion were happening at the citywide level” (page 18):
 - Suarez-Bertoa, R. et al. Primary emissions and secondary organic aerosol formation from the exhaust of a flex-fuel (ethanol) vehicle. *Atmospheric Environment* 117, 200-211, doi:<http://dx.doi.org/10.1016/j.atmosenv.2015.07.006> (2015).
 - Karjalainen, P. et al. Time-resolved characterization of primary particle emissions and secondary particle formation from a modern gasoline passenger car. *Atmos. Chem. Phys.* 16, 8559-8570, doi:10.5194/acp-16-8559-2016 (2016).
- In the context of “Monthly diesel wholesale shipments available only for the São Paulo state (including highway) market between 2008 and 2013 have gradually trended upward (Supplementary Figure S7)” (page 16):
 - “Over the study period, diesel prices broadly stayed constant in nominal terms, and gradually trended downward in inflation-adjusted terms (Supplementary Figure S5), so such variation is accounted for by site-specific time trends or year fixed effects that are included in our regression controls. Monthly diesel wholesale shipments available only for the São Paulo state (including highway) market between 2008 and 2013 have gradually trended upward (Supplementary Figure S7)⁵⁴”
 - Pérez-Martínez, P. J., de Fátima Andrade, M. & de Miranda, R. M. Traffic-related air quality trends in São Paulo, Brazil. *Journal of Geophysical Research: Atmospheres* 120, 6290-6304, doi:10.1002/2014JD022812 (2015).
- We do not cite the following two articles that Reviewer #2 asks us to review:
 - Ronkko et al. *Vehicle Engines Produce Exhaust Nanoparticles Even When Not Fueled* (2014).
Brief rationale: Engine braking conditions such as deceleration and downhill driving may contribute to particle emissions. We already correct for spatially and temporally resolved variation in traffic congestion in our regression models, and braking is unlikely to correlate with the gasoline-ethanol fuel mix.

- Liu et al. Impact of Vehicle Development and Fuel Quality on Exhaust Nanoparticle Emissions of Traffic (2013).

Brief rationale: Vehicle/engine development, for example, the introduction of gasoline direct injection (GDI), may be an important driver of particle emissions, but such developments in a real-world fleet should be gradual, accounted for by trends, and unlikely to fluctuate with the gasoline-ethanol mix over a relatively short and homogeneous time period that we examine.

• It seems there are many confounding effects on top of temperature, wind speed, boundary layer height, the spatial distribution of traffic, and even drifts. So, why using the econometric approach among others? You could use a different methodology, trying to identify sources and more focus on trace elements. Source distribution analysis of PM2.5/BC/element measurements from the researched sites with 24h filters could be performed using the Positive Matrix Factorization PMF of the Environmental Pollution Agency EPA for instance. At least comment other methodologies and possibilities.

We thank Reviewer #2 for this comment. We chose to adopt an empirical econometric approach, for the reasons outlined above, without implicating other approaches, which we view as complementary to our study and will be a valuable contribution to the literature. With respect to PMF, this would require physical access to the environmental authority's 24-hour filters, and we now mention source identification studies on page 5.

• Do they import sugarcane in São Paulo from India? I think this remark is not so important since harvest conditions in India likely do not affect Brazilian ethanol production neither international imports.

We clarify that the aim of the remark is to point out that ethanol price movements were the result of developments in world food and energy markets, rather than concerns over air pollution in São Paulo, which would make the main regressor of interest, the gasoline-ethanol mix, an "endogenous" variable (i.e., responding to the system we model, rather than "exogenous" to it). An expanded text is included in Supplementary Information, pages S2 and S7. In response to the Reviewer's comment, in the main text we drop "such as a poor sugarcane harvest in India in late 2009," and keep (these price fluctuations) "were driven by developments in world food and energy markets."

• Again, it is not complete that area attribute particle sizes below 100 nm are only related to direct emissions from light duty vehicles LDVs; other studies relate the emission of nanoparticles and bio-diesel and diesel consumption from trucking.

We have revised the text to reduce ambiguity, and are grateful to Reviewer #2 for inducing us to do so.

• *You mention that shifts from gasoline to ethanol use increased ozone concentrations. Increase of O₃ concentrations could also be related to NO_X emission changes from HDVs and subsequent diesel consumption.*¹¹

The ozone pollutant regression model reported in Fig. S8 now also includes ridership on diesel buses. While our manuscript's focus is on particles, we include a brief analysis of the effect of gasoline versus ethanol use on ozone, using an extended sample compared to the extant literature, with the purpose of illustrating the method. The ozone (and particle) analysis also controls for the inauguration of the southern section of the Greater São Paulo beltway, in April 2010, which reduced HDV traffic in the inner city, since Salvo and Geiger (2014) did not control for this explicitly (i.e., other than through the use of trends).

• *You mentioned a plausible scenario where ultrafines fell while PM_{2.5} remained invariant. Could also be possible another scenario: global decrease of PM_{2.5} concentrations simultaneous to nano-particles?*

We clarify that ultrafine levels did not fall; they fluctuated in tandem with the penetration of ethanol in the gasoline-ethanol light vehicle fleet.

Chapter Ultrafines rose with a shift to gasoline, and fell as consumers returned to ethanol:

• *Ultrafine variation was observed during mid-summer to mid-fall (January 20 to May 31) of 2011 (5 months). The period was shorter than the 11 month observations included in the previous section (5 months, Oct. 2010-sept 2011). I found confusing the timing of the different samplings, sites and species (be consistent throughout the manuscript). I would recommend including a summary table with all these data.*

As mentioned above, we follow the Reviewer's recommendation and include two summary tables. Table I summarizes the different datasets that our study combines. Table IV provides an overview of all the regression model specifications that we estimate.

On presenting the submicron particle regression estimates, we add some clarifying statements including, on page S24, "In addition to using the full sample that comprises three seasons (spring, summer, fall) and employing quarter-of-year fixed effects, we also consider a shorter subsample. In this alternative specification, we restrict the sample to the set of summer and fall months from late January to May 2011, during which seasonal variation is less pronounced and arguably "monotonic," and we also include a linear trend. Of note, this shorter sample still encompasses the second and most pronounced episode of ethanol price variation, marked by a rise followed by a drop in relative prices (SI Part A). The shorter sample also excludes the early January vacation period in which vehicle emissions may differ substantially. We refer to this shorter sample as the more seasonally homogeneous sample. In sum, our regression analysis uses both the full sample of measurements and, separately, the shorter mid-summer to fall sample in which seasonal variation is less pronounced."

We are grateful to Reviewer #2 for the opportunity to improve the communication of our work.

• *What does it mean a placebo test. Please omit subjective terminology. What about diesel/bio-diesel consumption rates?*

We no longer describe the test as a placebo. Our intention was to point out that one would expect black carbon concentrations to change with diesel consumption, not necessarily with gasoline-ethanol use.

• *In São Paulo there are studies observing a different fact on a long-term: overall increase of O₃ levels and decrease of NO_x concentrations, despite the increment of gasoline fuel use.¹¹ In a former study you explained the same evidence: towards O₃ reduction in the afternoon by the fuel shift from ethanol to diesel.¹² What about other side effects as NO_x increments from HDVs?*

We clarify that the article cited (Salvo and Geiger, 2014) examined how fluctuations in gasoline versus ethanol use – not diesel – in São Paulo affected ozone levels. As is the case in the present manuscript, in that study diesel consumption did not vary beyond a trend, i.e., diesel use did not fluctuate in tandem with the gasoline-ethanol mix.

Chapter Tight co-variation during morning rush hour:

• *Ultrafine concentrations due to increased gasoline in the early morning. During early morning weather conditions are more stable due to lower planet boundary layer (PBL) and lower wind speeds. Weather dispersion conditions could have worsened and/or bio-diesel consumption could have changed during the researched five months, masking your results. If fuel shift would be the main driving reason, I would also expect evening significant UFP increases during peak hours where weather conditions are unstable and may have not change much during these months. Do you have any explanation for that? In other studies it was found a different behavior with heavy-duty vehicle (HDV) traffic decreases: lower pollutant concentrations during the Evening Peak compared to the Morning Peak.*

Our regression models control for PBL and wind conditions.

Hour of the day is a key predictor of ambient particle levels, for anthropogenic and natural reasons, and this is why our analysis examines variation by time of the day separately.

Please see the preceding comments on the low penetration of biodiesel and how we control for variation in diesel over the sample period.

Consistent with Reviewer #3's last sentence, in all our one-hour datasets, particle levels in the morning peak exceed those in the evening peak. This can be seen in Figs. S12b (aerosol particle number concentration), S13b (BC), and S14b (PM_{2.5} up to 22:00), shown in the original submission. Again, the task of our empirical approach is to uncover the *variation* caused by gasoline-ethanol transitions, not to explain background levels of particles.

Chapter Variation in 24-hours means:

• *Estimates from model with the ethanol-to-gasoline price ratio serving variable for the predicted gasoline share. Why not using fuel sale statistics on a daily basis instead of this proxy*

based on fuel prizes? What about obtaining fuel sales from fiscal data? Why using ethanol/gasoline relative prizes on a multinomial probit choice model? Sometimes users are reluctant to change fuel (inelastic demand instead of less elastic demand). How do you establish background levels of emissions that were unlikely to vary with the ethanol price fluctuations that occurred over the space of months? The explanation given in the text is confusing.

Per the preceding comments, fuel quantities on a daily basis for the São Paulo metropolis are not publicly available. The fuel quantity data that are available are wholesale shipments at the monthly level for the entire state of São Paulo, data that we use to form an alternative proxy for the gasoline share (denoted $s_t^{gas,aggr}$). Our findings are robust to this sensitivity analysis, which can be seen by comparing columns marked “Share in the aggregate fleet” to columns “Share in the flex fleet” in, for example, Tables SII (24-hour PM2.5, filter measurements), SIII (morning PM2.5, beta measurements), SIV (ozone as an illustration of the method).

We note from page 17 (and we now add a similar but shorter comment on page 4): “The reason we need a demand model to predict day-to-day fuel quantities from day-to-day fuel prices is that high-frequency fuel quantity (usage) data for the São Paulo metropolis are not available, only price data. Otherwise, we would skip the first-step model and use the fuel quantity data directly in the second step.”

The estimated consumer demand system, based on a multinomial probit choice model, agrees with Reviewer #2’s description that “*Sometimes users are reluctant to change fuel (inelastic demand instead of less elastic demand)*”. We state on page S8: “In particular, these empirical demand studies report that consumer preferences and behavior are such that flex-fuel vehicle drivers (overwhelmingly household consumers) do not as a whole transition abruptly between gasoline and ethanol at – or even close to – the relative price point at which the effective prices of ethanol and gasoline, in \$/km of distance traveled, are equalized. Instead, consumer switching is significantly more gradual – demand is less “elastic” – around this parity price ratio or threshold, that lies just under 70% for most vehicle models, $p_e/p_g \approx 0.70$. To further describe consumer substitution patterns...” (the text continues until) “The approximately linear demand relationship between fuel shares and the fuel price ratio is also shown in Salvo and Geiger (2014), Fig. 3A.”

Chapter Discussion:

- *Absence of a significant relationship outside this time window and size range. What about changing driving conditions during the observed months (maybe you have more congestion during morning peak hours).*

We clarify that this is precisely the reason why all our regression models control for citywide and local road traffic congestion. To the paragraph, we now add: “The potential confounding factors that we control for include a trend and meteorological and road traffic conditions recorded concurrent to the day and hour.”

- *Laboratory experiments cited earlier. Which ones?*

We have now edited the text to improve clarity, citing again five specific references that we had cited in the introduction.

• *Insignificant association between the light-vehicle fuel mix and BC levels. Maybe the sources are different?*

The main contributor to BC in the metropolis is the HDV fleet, as Reviewer #2 indeed has suggested. This was stated in the Introduction and in the subsequent paragraph. At this point of the Discussion, we now state: "...insignificant association between the light-vehicle gasoline-ethanol mix and BC levels, which are influenced mainly by diesel combustion in heavy-duty vehicles^{39,43,44}. Fifth, controlling for monthly ridership of diesel buses in the metropolis does not change our estimates, due to bus ridership not varying over the sample period (SI Part D)."

References

1. *Atmospheric Environment* 117 (2015) 200-211
2. *Fuel Processing Technology* 96 (2012) 237-249
3. *Science of the Total Environment* 409 (2011) 738-747
4. *Fuel* 134 (2014) 201-208
5. *Science of the Total Environment* 500-501 (2014) 64-71
6. *Front. Environ. Sci.* 3:82. doi: 10.3389/fenvs.2015.00082
7. *Environ Sci Technol.* 2014;48(3):2043-50. doi: 10.1021/es405687m.
8. *Environ. Sci. Technol.* 2013, 47, 8091-8092. dx.doi.org/10.1021/es401805r
9. *Atmos. Chem. Phys.*, 16, 8559-8570, 2016. doi:10.5194/acp-16-8559-2016
10. *Atmospheric Environment* 107 (2015) 374e390
11. *Journal of Geophysical Research: Atmospheres* 120 (12), 6290-6304
12. *Nature Geoscience* 7, 450-458, doi:10.1038/ngeo2144 (2014).

Reproduction of new Fig. S8, to address Reviewer #2's concern regarding the role of diesel combustion as a possible confounder in the analysis.

Sensitivity analysis, new Fig. S8: Adding a control for monthly diesel bus ridership in the public transport system (the variable is plotted in Fig. S6) to all the regression models:

Original Fig. 1 in the main text:

Reproduction of new Supplementary Information Part D and accompanying figures, to address Reviewer #2’s concern regarding the role of diesel combustion as a possible confounder in the analysis.

D. Diesel Combustion in the Heavy-Duty Vehicle Fleet. We argue that diesel combustion in heavy vehicles, while an important contributor to particle emissions and secondary particle formation, is unlikely to confound our inference of the effect on particles of gasoline versus ethanol use in light vehicles during the periods we examine. Fig. S5 reports variation in the retail price of diesel oil in the São Paulo metropolitan area between November 2008 and May 2013 (the longest period of PM_{2.5} monitoring). The figure plots the monthly consumer price index (base price index in October 2008 = 100) for diesel and also, for comparison, the price indices for gasoline and ethanol, both in nominal (inflation-unadjusted) and real (inflation-adjusted) terms.

After a downward 5% price adjustment in mid 2009, diesel prices stayed constant in nominal terms, and gradually declined in real terms, until mid 2012, when the federal government began to partially adjust diesel prices for cumulative inflation observed in the preceding years. In real terms, diesel prices in May 2013 were still below their October 2008 level, as was the case for gasoline prices. In particular, diesel prices hardly changed in nominal terms (and hardly changed beyond a gradual downward trend in real terms) during the DMPS sampling campaign that ran from October 2010 to September 2011.

In contrast to pronounced fluctuations in the price of ethanol, the subdued variation in the price of diesel – including the absence of fluctuations – suggests that diesel use is unlikely to be a confounder in our regression analysis. If anything, diesel prices in real terms followed a gradual downward trend over several years, and our particle regression models include a time trend, which absorbs the effect of any omitted determinant of particle concentrations that exhibits a trend. Controlling for diesel prices in the particle regression, as we do in robustness tests, indeed does not change our results.

We provide two additional pieces of evidence to underscore the point that “omitted variable bias” due to variation in diesel combustion is unlikely to be present. First, buses in the public transport system are a key source of diesel emissions in the São Paulo metropolitan area. Fig. S6 reports monthly ridership on buses in the public transport system from November 2008 to May 2013. (We also show the evolution of the ethanol-to-gasoline price ratio.) Ridership was quite stable over the period, tending to fall in the month of January due to the yearend school vacation period, and similarly in the winter month of July in which schools also break (these days are either controlled for using separate type-of-day fixed effects, or excluded from our regression samples). There is no indication that commuting on (use of) diesel buses responded to the gradual decline in real diesel prices (Fig. S5), consistent with the provision of public transport being insensitive to diesel prices (which hardly varied in the first place). Moreover, there is no indication that flex-fuel vehicle motorists might have taken to public transport as ethanol prices rose, which could otherwise confound our inference (SI Part F). Controlling for diesel bus ridership in the particle regression, as we do in robustness tests, does not change our results.

Second, we obtained monthly diesel fuel shipments reported by wholesalers for the state of São Paulo – the same data source (ANP) as the wholesale gasoline and ethanol fuel shipments from which we compute $s_t^{gas,agg}$ for use in sensitivity analysis (SI Parts C and G). Unfortunately, these diesel shipments include the large and seasonal statewide highway market;

diesel volumes specific to the São Paulo metropolitan market are not publicly available. Fig. S7 shows that state-level diesel shipments over the course of the first semester of 2011 broadly followed their typical upward seasonal trend. Importantly, there is no evidence of confounding correlation with the pronounced fluctuation in the price of ethanol relative to gasoline and in the gasoline share – namely up until the beginning of April 2011, and down thereafter – that we exploit in our empirical analysis (noting, again, that our regression models control for trends).

For completeness, we note the following changes in the composition of diesel during the full sample period. These changes, mandated nationwide by the federal government, slightly increased the proportion of biodiesel added to diesel oil: (i) from 3% to 4% beginning July 2009; (ii) and from 4% to 5% beginning January 2010.

In sum, the evidence indicates that potentially confounding effects on the particle size distribution from variation in heavy-duty vehicle traffic around the time of each ethanol price hike are unlikely.

Fig. S5. Monthly price indices for diesel oil (dashed black line), regular-grade gasoline (E25 or E20, crossed red line), and regular-grade ethanol (E100, thick green line), at the pump in the São Paulo metropolitan area, from October 2008 to May 2013. In the top panel, price indices are deflated to account for variation in the economy-wide price level (IPCA Brasil). In the bottom panel, price indices are not adjusted for inflation. Base October 2008 = 100. Source: IBGE.

Fig. S6. Ridership on diesel buses in the public transport system in the São Paulo metropolitan area (left vertical axis) and average ethanol-to-gasoline price ratio across the city's pumps (dashed green line, right vertical axis), by month from November 2008 to May 2013. Average daily rates of ridership in millions of passengers per day are reported by dividing the month's total ridership by the number of calendar days in the month. Sources: SPTrans (Prefeitura de São Paulo, Transportes), ANP.

Fig. S7. Monthly wholesale diesel fuel shipments for the state of São Paulo, including the state-wide highway market, over the first semester of 2011, compared to the first semester in other years. During semester 1, 2011, diesel prices hardly varied whereas ethanol prices fluctuated markedly relative to gasoline prices (Fig. S5). Sources: ANP.

Reviewer #3 (Remarks to the Author):

This is an interesting article attempting to link the shift of gasoline use to ethanol with the decreased ultrafine particles. There are many studies on the emissions of ultrafine particles from gasoline and ethanol combustion.

We appreciate the Reviewer's comments and highlight that while there are a number of studies assessing fuel choice in flames and engines on nanoparticles, this work is the first to do so for an urban area subject to actual – as opposed to hypothetical – changes in ethanol vs gasoline consumption. Our work is timely, given that a review as recent as 2014 by Kumar et al., cited by the reviewer and now included in our manuscript revision, points out that “the effects of alternative fuels, such as biofuels, on UFP emissions are yet to be understood.”

We now cite the review recommended by the Reviewer as early as our opening paragraph.

This study assess the concentrations to reach to conclusion that that this shift show a decrease in ambient concentrations. There are a number of issues and the findings can be challenged. For example, the assessment is made based on the measurements on a limited location which can not be taken as representative of the complex cities like Sao Paulo. Moreover, ultrafine particles are short-lived particles meaning that the sites away from the sources will not be representing the real emissions from the road traffic as they will be subjected to transformation processes such as nucleation, coagulation and condensation.

We certainly appreciate this point, which is why we included, in the Supplementary Information Part B of the original submission, our findings of highly correlated times series for PM_{2.5} and BC mass concentrations across sampling sites and methods over days in the sample periods. The relevance of this finding may not be necessarily applicable to ultrafines, of course. Instead, the location of DMPS instrument at USP very likely provides a lower ultrafine particle number concentration when compared to a roadside location, or measurements in a vehicle's plume.

We have therefore added the following statement to the main text on page 14 (and provide additional citations): “The location of this instrument at the USP campus very likely provides a lower ultrafine particle number concentration when compared to a roadside or on-road location, suggesting that the absolute changes in particle number concentrations that coincide with the fuel shifts may be different if they were to be measured near road traffic or at the vehicle exhaust^{50,51}.”

We also reproduce here our response to Reviewer #2, who mentioned the sampling at one site over a one-year period:

It is also important to note that “quasi-experimental” variation of this scale, in which two million actively circulating passenger cars transitioned from ethanol to gasoline – and then returned to ethanol – within the space of a few months, is rare. The fact that the price of ethanol relative to gasoline has hardly varied since 2011, all the way to the present day, 2016, illustrates this. (Figs. S4 and S5 show this lack of price variation between 2011 and 2013.)

Reviewer #3 also states that additional sampling sites would be beneficial. Such additional instrumentation was not available for use by the authors and, in the case of a real-world episode at the scale of a megacity, more instruments cannot be deployed retroactively. Kumar et al. (2014), the review cited by Reviewer #3, characterize the instrumentation and measurement technology for ultrafine particles by their “high cost for field deployment in sufficient numbers” as well as “lack of robustness for long-term unattended operation” (p.7). The rooftop of a four-storey building inside the USP Armando Salles de Oliveira campus was chosen due to the relatively well mixed urban air, with influence from representative road emissions sources but not excessive proximity to idiosyncratic emissions. Two of the co-authors have used the site extensively in previous research. For example, Backman et al. chose this same site, describing it as follows on p.11735:

“The measurements were made at the Armando Salles de Oliveira campus area of USP (Fig. 1). The campus area is a vast park, totalling an area of 7.4 km², **making the site ideal for tracking ambient aerosols, without strong local sources**. At the campus area buildings are scarce. Thus, air masses arriving at the station should be well mixed and make the measurements **representative of the ambient pollution burden of the city.**”

(On the diurnal cycle of urban aerosols, black carbon and the occurrence of new particle formation events in springtime São Paulo, Brazil, J. Backman, L. V. Rizzo, J. Hakala, T. Nieminen, H. E. Manninen, F. Morais, P. P. Aalto, E. Siivola, S. Carbone, R. Hillamo, P. Artaxo, A. Virkkula, T. Petaja1, and M. Kulmala, Atmos. Chem. Phys., 12, 11733–11751, 2012)

Moreover, the biggest challenge is that a difference of 8000 particles per cm⁻³ (unit value seems wrong on fig 1 - it should be perhaps 10³ (not 10⁻³)...

We thank the Reviewer for his/her careful reading, and have changed cm⁻³ × 10⁻³ to cm⁻³ × 10³.

...is taken to conclude the effect of this shift. The urban concentrations has much more variability in the concentrations (of the order of 10⁴ cm⁻³, meaning that this difference cannot be taken as a direct effect of shift in fuel and many factors could have contributed to this change (e.g. distance of the measurement from the road, size range considered;...

We note that our analysis controls for – that is, filters out – many determinants of the particle size distribution that might add “noise” to raw data (measurements). These controls are either:

- (i) by experimental design, e.g., we do not vary the distance of measurement from roads;
- (ii) by estimation sample, e.g., we examine effects separately by hour of day; for weekdays only given that commuting patterns differ on weekends and holidays; and we exclude the colder months of June to September (the marked ethanol price variation occurred outside these months);
- (iii) by considering, as our preferred specification, a sample that is relatively short and during which the fuel mix fluctuated markedly in different directions (more gasoline,

- then less gasoline), so that the co-variation that we uncover is not estimated off a trend, and is unlikely to suffer from omitted variable bias. The sample is quite homogeneous other than for the variation in the gasoline-ethanol fuel mix, our variable of interest. As sensitivity analysis, we use a longer sample, starting in the spring/October rather than starting mid summer/late January (to mid fall); or
- (iv) by including regression controls, importantly, meteorological (e.g., wind speed, precipitation), PBL and road traffic conditions recorded concurrent to the particle measurements.

As we note on page 8, we make these choices “with a view to limiting unobserved determinants of the particle size distribution, which might bias our estimates of the effect of the fuel mix or make them less precise.”

We respectfully refer to Prof. Sasha Madronich, who in his piece accompanying Salvo and Geiger (2014)’s ozone study wrote: “Establishing whether changes in pollutant levels could be related to shifts in fuel use was not a simple task: the influence of weekly, seasonal and longer term variations in pollutant levels, and key meteorological variables, had to be taken into account. **Once these factors had been corrected for, however, it became clear** that a shift from ethanol towards gasoline use was accompanied by a decrease in ground-level ozone concentrations” (emphasis added).

If our understanding is correct, Reviewer #3 is concerned that a change in 8700 particles per cm³ (Table I, UFP 7-100 nm, 08:00) may be difficult to detect in an environment that exhibits highly variable particle size distributions. We emphasize that this is precisely why the econometric method, coupled with a design that is sensitive to correcting for determinants of particle levels *other* than those that are the object of study, is powerful.

The econometric method (statistics that takes into account human behavior, such as drivers responding to variables such as fuel prices) fully recognizes the importance of having to account for variability in data, which is the rationale for using the method in the first place. Critically, if fluctuations in ultrafine particle levels on the order of several hundred percent were to happen for “unobservable” reasons, other than due to determinants such as meteorology and seasonality that we explicitly account for, this would be reflected in large confidence intervals on our estimated effects, which would prevent us from making statistical inference. In contrast, our results for UFP 7-100 nm in the morning peak are statistically significant, indicating that our ultrafine measurements are not “noisy” (i.e., our results feature a high “signal” to “noise” ratio, or high explanatory power, R²).

Only to provide an analogous example from our daily lives, imagine that in typical classrooms, student scores may vary between 15% and 95%. A university seeks to infer the effect of an intervention – say, a different teaching method – on average student scores. Finding that the intervention has a statistically significant effect of +5 percentage points on average student scores (say with 95% CI of [3,7] percentage points) would be deemed very policy relevant. This conclusion holds irrespective of the fact that much variation in student scores remains unexplained, namely that the range in student ability is of the order of 95-15=80%, or 16

times the average effect of the intervention (5%). In this sense, the importance of the 8700 cm^{-3} shift that we find is not undone by the fact that many other determinants shift ultrafine particle levels. The result of $+8700$ particles per cm^{-3} shifts the average particle size distribution by that amount. We hope that our analogy helps clarify rather than confuse.

Finally, to provide added perspective, we should point out that the one-hour readings for UFP 7-100 nm in our raw data do *not* exhibit greater variability when compared to the same-hour readings for PM2.5 at the single monitoring site in the metropolis where the environmental authority took official measurements over an overlapping period. Consider, for example, non-holiday weekdays between January 20, 2011 and May 31, 2011 (this is the more seasonally homogeneous sample used to generate the submicron results in Table II and Figs. 1 to 3). Specifically, the two box plots below show medians and interquartile ranges (p75 – p25 shown in the thick bands), by hour of day, in the UFP 7-100 nm (cm^{-3}) and PM2.5 ($\mu\text{g}/\text{m}^3$) samples:

(a) One-hour UFP 7-100 nm (cm^{-3}), at a single site (USP campus) in the SP metropolis, January 20, 2011 to May 31, 2011

(b) One-hour PM2.5 ($\mu\text{g}/\text{m}^3$), at a single site (Congonhas) in the SP metropolis, January 20, 2011 to May 31, 2011. Source: CETESB

Importantly, in plot (a), for a given hour, the UFP 7-100 nm interquartile range is about 7900 cm^{-3} , or 56% of the size of the mean (14100 cm^{-3} , also averaging across hours). In plot (b), the PM_{2.5} interquartile range is about $14 \mu\text{g}/\text{m}^3$, or 61% of the size of the mean ($23 \mu\text{g}/\text{m}^3$).

Judged against another particle size range and physical parameter, the variability in our data of one-hour mean UFP 7-100 nm is not “out of the ordinary.”

Two additional comments are in order. First, UFP 7-100 nm variability exhibits a more pronounced diurnal cycle, which is another reason why we implement our regressions hour by hour – recall item (ii) above. Second, variability in the raw data shrinks as we progressively add explanatory variables to the regression equation, e.g., the factors listed in item (iv) above, in addition to the main explanatory variable of interest, the gasoline share. But this variability will not shrink to zero, as residual variation – the econometric error – always remains.)

We have now added the following statements to:

- (i) the main text on page 4: “In the second step, the econometric/statistical approach corrects for³⁴⁻³⁶ potentially high variability in nanoparticle levels. Specifically, the analysis fixes or controls for typical factors of nanoparticle variation^{8,10}, including the distance of measurement from roads, the time of day, the day of the week, seasonality and longer term trends, key meteorological variables, traffic congestion, and the combustion of fuels other than gasoline and ethanol, which are our object of interest. The econometric approach requires that careful consideration be given to whether remaining, unobservable determinants of nanoparticles might co-vary with the gasoline-ethanol mix, and the evidence suggests not (see Methods below).”
- (ii) the text that accompanies Fig. S15, which provides a graphical analysis of residuals (Supplementary Information page S31): “Even in this more seasonally homogeneous sample – notice that there is no obvious trend – and fixing the hour of the reading (08:00, one-hour average), there is substantial variability in the raw data (much as there is variability in the raw 24-hour PM_{2.5} data in Fig. S3). For example, the 7-100 nm values range between under $10,000 \text{ cm}^{-3}$ to almost $40,000 \text{ cm}^{-3}$.”

... instruments (different instruments such as CPC and DMPS are used without harmonising their data as these instruments could provide different results while measuring the same level of concentrations)...

We thank Reviewer #3 for requesting this clarification. We have now added the following text to Supplementary Information Part A, “Particle Sampling: Methods, Sites and Temporal Coverage”:

“We validated the DMPS measurements against an independent condensation particle counter (CPC, model 3022, TSI Inc., St. Paul, MN, USA), operated concurrently to the DMPS and at a similar lower size cut. As a result of the data validation, less than 2% of the original DMPS data were removed due to a deviation of the integrated aerosol number concentration 50% or higher than the aerosol number concentration measured independently by the CPC. A linear fit

between the DMPS integrated concentration and the CPC concentration yields an R2 of 0.99, with a slope of 1.18 (Fig. S2a). The diurnal variation (median) and variance (interquartile range) of both measurements shows very tight correlation, without differential trends throughout the day (Fig. S2b).”

For convenience, we reproduce Fig. S2 at the end of our response to Reviewer #3. We have also edited the Methods section accordingly.

... and the dispersion conditions led by the atmospheric stability conditions.

Please see the preceding comments.

A number of directly relevant papers are overlooked which could have helped to build the arguments (e.g., Ultrafine particles in Cities, Environmental International) and enrich the discussions. While I see this an interesting study, the article is not fit for journals like Nature due to a number of weak points.

We are grateful to Reviewer for pointing out the Kumar et al. (2014) review, which indeed makes arguments and points out needs that are being addressed here. We note that the only South American city surveyed in that review is Santiago de Chile, which is home to half the population of São Paulo and subject to significantly different orographic features and meteorology (precipitation, temperature) when compared to the São Paulo metropolitan area.

The results presented in our manuscript are twofold: (1) we are able to quantify the reduction in primary aerosol emissions from a large-scale real-world shift from gasoline to ethanol use (~30%, concentrated in the <100nm range); and (2) there is little indication of a significant change in SOA formation rates, as the reduction due to the shift in the fuel mix is significant only for the morning rush hours. We view this contribution as appealing to a broad audience given the expectation that biofuels may play an important role as nations invest in climate change mitigation, e.g., the US looking to penetrate the E10 “blend wall” by adopting E15 gasoline. Importantly, by providing the first observational evidence, our contribution is hopefully to stimulate subsequent research by pointing out the next areas to investigate, precisely to call for more research, including elucidating the mechanism, modeling, and chemical analysis of the nanoparticles.

Reproduction of new Fig. S2 on harmonization between DMPS and CPC, in response to Reviewer #3's concern that the two instruments might be used without harmonizing their data.

Fig. S2. Harmonization between DMPS and CPC. The panels compare aerosol number concentrations integrated from the DMPS and from an independent CPC. a, Scatterplot and line of best fit between values in both datasets. b, Diurnal variation (median, marked by the thick lines) and variability (the interquartile range is shaded) in each dataset.

Reviewers' Comments:

Reviewer #2 (Remarks to the Author)

First of all, I would like to thank the authors for the rigorous effort and discipline to explain some of the research questions presented. Most of the points discussed in the previous review process were carefully addressed by the authors in this new version. Now, the authors complement the study carefully with additional literature dealing with different approaches and analysis techniques. However, the main discussion addressed in the previous review, "explanation of an universal law from a case study", was unfortunately neglected.

The main concern is related to the adequacy of the sampling spot for the interests of the research. The authors elucidate a-priori conclusions and the hypotheses based on a non-traffic/background sampling plot could be wrong. The plot is either far from the main vehicular sources and/or is affected by other non anthropogenic local sources (the site is heterogeneous and has different sources even biological ones). If there were other causes and impacts driving to paper's a-priori conclusions - as purchase of new public vehicles or use of new diesel fuels and sampling errors/resolutions in the instruments - they were completely neglected. What about resolution of the sampling instrument? I do not know how this econometric deterministic model can deal with this constraint. Some of the explanations given in this sense are contradictory. You even accept that the location was not suitable for this kind of analysis, does not reproduce real world traffic conditions and the results could not be extrapolated to the entire Metropolitan Region (two million vehicles did not passed through the University Campus, obviously). We were not asking for a full network of plots but at least you would need a second plot - more representative of real world driving conditions - to confirm your main hypothesis. The sampling site could be not appropriate for the research's interests and the definition of the experiment could be improved substantially. The main road highways are distant from the sampling spot.

Although the authors now include new variables in the econometric model, the main concerns discussed before were not correctly explained. For instance, despite passenger ridership and fuel consumption increments, the technology of diesel vehicles could have changed considerably during last 5 years (do not use riderships but vehicle technology). The problem of using such a deterministic model is that exclusively depends on the initial variables considered, ignoring other processes and knowing "almost nothing" about the real mechanisms behind the increases or decreases of the nano-particles emissions (working as a black box). You could have used meteorological parameters, vehicle fleet removal rates or data on local traffic changes instead and the results would had been completely different. It was the reason to cite at least other approaches and techniques.

There are also some conceptual topics that must be clarified. In my understanding, you do not have trucks entering the university campus but bus lines are used for commuting. Buses are a key source of diesel emissions in São Paulo, as you mentioned, and maybe traffic conditions related to bus lines have changed during this period. Instead of using regional aggregate statistics on fuel consumption you could have used local transit data from companies (i.e frequencies/headways of bus lines). I wonder how State's diesel prize policies could have affected the decrease of nano-particles at this sampling site. Once again, a redefinition of the research hypothesis and experiments should be done. The same for the other transport related fuel policies. Regarding O₃ formation, how can ridership affect O₃ formation changes? There must be another causes of this O₃ increments.

Despite of my comments I encourage the authors to continue with this publication focussing the research topics more on a case study, with innovative advances from preliminary results, instead of an impact and ambitious "universal law", really difficult to demonstrate and prove at this state.

Reviewer #3 (Remarks to the Author)

The authors have done well to respond to individual comments as comprehensively as possible. However, the inherent weaknesses of the paper remain in the form of drawing conclusions based on limited stations and using the rooftop data to make conclusions for ultrafine particles that are short-lived and not a real representation of the on-road scenario. The authors have accepted these limitations and therefore leaving on the editor to make a final call whether a work with such limitations is accepted by the journal.

Reviewers' comments (in italics):

Reviewer #2 (Remarks to the Author):

First of all, I would like to thank the authors for the rigorous effort and discipline to explain some of the research questions presented. Most of the points discussed in the previous review process were carefully addressed by the authors in this new version. Now, the authors complement the study carefully with additional literature dealing with different approaches and analysis techniques.

[Reviewer's concluding statement] *Despite of my comments I encourage the authors to continue with this publication focussing the research topics more on a case study, with innovative advances from preliminary results, instead of an impact and ambitious "universal law", really difficult to demonstrate and prove at this state.*

We very much appreciate and thank the Reviewer for his/her effort to improve our work. We are hopeful the responses below address any remaining concerns.

However, the main discussion addressed in the previous review, "explanation of an universal law from a case study", was unfortunately neglected.

Before going into the detailed responses, we wish to clarify that we do not claim an "explanation of an universal law from a case study." Instead, the manuscript presents an empirical study (or, as the Reviewer refers to it, a case study¹) that is accompanied by appropriately chosen caveats and words of caution throughout the manuscript. Indeed, we have reread the manuscript carefully, editing at selected locations – including the abstract – with this purpose in mind. Below, we provide detailed responses to the valuable comments made by the Reviewer.

1. Our abstract, already in the previous version, cautions: "The finding motivates further studies in real-world environments." Similar words are found on p.12 ("... would be an important next step...") and on p.13 ("...motivates further studies..."). We have now added more words of caution to our concluding paragraph: "As with any empirical observational study, confidence in its findings can only grow as new samples, in space and time, become available, supported by the results from different approaches and analysis techniques." We emphasize that our sample, based on a real-world experiment at this scale, is not easily reproducible. In particular, since 2011 to the present date, the price of ethanol relative to gasoline has hardly fluctuated (see Figs. S4 and S5 for relative price fluctuation in the 2008-2013 period), and it is this experimental "lever" that induces a large fraction of the light-vehicle fleet to switch between fuels. We are fortunate to have had ultrafine particle sampling instrumentation deployed in the field – on the runway, so to speak – back in the first semester of 2011, when relative

¹ To case study, we should add that it is a case study of the *single* episode of gasoline-ethanol fuel mix variation at this scale ever observed anywhere to date. This is important to bear in mind in the discussion surrounding paragraphs 1 and 6 below.

prices last fluctuated widely. We are also fortunate that, over this same period, as suggested by the evidence, other determinants of ultrafine particle levels such as diesel combustion did *not* fluctuate (i.e., shift in different directions) in tandem with relative ethanol-gasoline prices (diesel quantity and price variation are shown in Figs. S5-S7). As mentioned in p.20 of our earlier response to Reviewer #3, on citing Kumar et al. (2014), DMPS instrumentation is costly to deploy, and our ultrafine sampling size was carefully chosen, building on previous published studies, a point we turn to next.

The main concern is related to the adequacy of the sampling spot for the interests of the research. The authors elucidate a-priori conclusions and the hypotheses based on a non-traffic/background sampling plot could be wrong. The plot is either far from the main vehicular sources and/or is affected by other non anthropogenic local sources (the site is heterogeneous and has different sources even biological ones). If there were other causes and impacts driving to paper's a-priori conclusions - as purchase of new public vehicles or use of new diesel fuels and sampling errors/resolutions in the instruments - they were completely neglected. What about resolution of the sampling instrument? I do not know how this econometric deterministic model can deal with this constraint. Some of the explanations given in this sense are contradictory. You even accept that the location was not suitable for this kind of analysis, does not reproduce real world traffic conditions and the results could not be extrapolated to the entire Metropolitan Region (two million vehicles did not passed through the University Campus, obviously). We were not asking for a full network of plots but at least you would need a second plot - more representative of real world driving conditions – to confirm your main hypothesis. The sampling site could be not appropriate for the research's interests and the definition of the experiment could be improved substantially. The main road highways are distant from the sampling spot.

The Reviewer makes several related points, to which we respond roughly in order:

2. The ultrafine particle sampling site was very carefully chosen. It is a “tried and tested” stationary site. With 6 million passenger cars and 0.3 million heavy-duty vehicles circulating and emitting across the metropolis, it is critical to choose a site that is representative for the whole urban area. Critically, the air is well mixed—it is neither a site that is influenced by one passing accelerating smoky vehicle (or placed in its plume) or idiosyncratic construction next door, nor is it a background site. Importantly, the site lies at a 1 km radius from a major road corridor (Marginal Pinheiros, running northwest to south, spanning over 200 degrees, and 20 express/local lanes in two directions) and is surrounded by busy roads (e.g., Corifeu de Azevedo Marques to the west-southwest). This can be seen in Fig. S1.
3. On p.20 of our earlier response to Reviewer #3, we provided a quote from Backman et al. (2012), who used the same site, on the adequacy—optimality, if we may—of the site: “site ideal for tracking ambient aerosols, without strong local sources” and “representative of the ambient pollution burden of the city.” (Backman et al. was a large team of researchers with extensive experience handling such instrumentation in the field: they chose against the “on-road scenario.”) We also noted in our earlier response to Reviewer #3, citing the Kumar et al. (2014) review, the technology’s “high cost for field deployment.” Our hope is that our research will motivate more measurements over time and in space.

-
4. We have now added much of paragraph 2, on p.15, immediately after the Backman et al. (2012) quote that we had added earlier when we responded to Reviewer #3.
 5. The two “non-anthropogenic local sources” we can think of are (i) potential biogenic VOC emissions from trees at the University of São Paulo campus, and ii) occasional sea salt intrusions over the city area, located at about a 70 km distance and 800 meter altitude from the coast. The green space on the USP campus is characterized by generously spaced trees, which, collectively, represent a small source of emissions compared to the dense vehicular traffic from nearby road corridors (containing up to twenty lanes) that completely surrounds the area and emits into the well-mixed air masses arriving at the measurement station. Given that the vegetated area is small, that there is little foliage in the city, we argue that biogenic emissions were unlikely to vary over the five-month sampling period in order to change nanoparticle concentrations (and in any case they would tend to be more pronounced in the afternoon and at night, not during the morning rush hour). Yet, we caution that VOC speciation is unfortunately not available for the sampling period, as a PTR-MS instrument was installed only in 2012. With regard to sea salt intrusions, these are mainly associated with coarse mode particles that do not affect ambient nanoparticle levels at the megacity’s distance and altitude from the coast. To p.15, we have now added the brief statement: “Non-anthropogenic influences, particularly in the ultrafine range, are limited. For example, vegetation on campus is limited compared to the dense vehicular traffic flows that surround it.”
 6. We clarify that a discussion of sampling errors/resolutions in the instruments can be found, in the earlier version, in Supplementary Information (SI) Part A, including Fig. S2, “Harmonization between DMPS and CPC.” The subsection “Particle sampling” in Methods links to SI Part A, and on p.14, row 14 states that DMPS measurements were validated against an independently operated CPC. We now also explicitly state, on SI p.S3: “Aerosol and sheath flow for the DMPS, CPC setup against an electrometer, compensation for system diffusion losses, and all other calibrations, adjustments and maintenance procedures follow Backman et al. (2012) exactly, and can be found there.”
[Editor’s comment] (on “you would need a second plot”) *this in particular needs to be addressed. e.g. why is this not possible? are there limitations to just having one site? would it make a difference to results/conclusions? - if potentially yes then a caveat needs to be added to your conclusions*

Deploying one set of instruments at an optimally chosen site was the best we could manage – and arguably the reasonable course of action given our knowledge at that point in time. A second measurement site was not in operation, deployed by us or by other researchers, during the periods of varying fuel mix we studied. Without going back in time, or waiting for a new period of varying fuel mix, there is unfortunately no option of adding a second measurement site. For the reasons argued above, our conclusions based on ultrafine particle sampling at the USP site are representative for the urban area. Of course, as with any empirical study, further sampling in time and/or space will tighten our inference and strengthen our results (see earlier and added words of caution, per paragraph 1).

Although the authors now include new variables in the econometric model, the main concerns discussed before were not correctly explained. For instance, despite passenger

ridership and fuel consumption increments, the technology of diesel vehicles could have changed considerably during last 5 years (do not use riderships but vehicle technology).

7. To clarify, and to support our main result on ultrafine particles – shown in column (3) of Table II and Figs. 1 to 3 – we do not examine a 5-year sample period, rather, we examine, as our preferred sample, a seasonally homogeneous period of just under 5 months, from January to May 2011.

Over the ultrafine-particle sampling period of less than one year, vehicle technology and, relatedly, diesel fuel composition (note we listed changes to fuel composition on p.18, row 5 in response to first-round comments), and age/maintenance of diesel vehicles did not fluctuate (i.e., shift first in one, then in the opposite, direction) (CETESB 2011 and adjacent years. Moreover, given its stock nature, fleet aging and turnover typically trends, rather than exhibit jumps, and our models account for trends (meaning that estimates are not based off trends). Specifically, any confounder to our result would have to reverse midway through this time window, e.g., more polluting diesel vehicles would be introduced gradually beginning mid January 2011, but then removed gradually beginning early April through the end of May 2011 (see Figure 2d). Such a confounder would violate the “identifying assumptions” that we specify in expressions [2] to [4] on pp. 20-23 in the Methods section, and discuss further in paragraph 9 below. There is no evidence to support such a hypothesis. This is why we added, in response to the Reviewer’s first-round comments, a control for ridership in the public transportation system, to demonstrate that (for example), ridership on diesel buses did not grow from January to March 2011 then shrink from March to May 2011.

[Editor’s comment] *This needs to be made very clear as in your manuscript this is a little confusing with the following in the manuscript methods as I cannot find anywhere you explicitly say 5 months, instead you state 5 years, 2 years and 11 months as sampling times. This needs to be clarified and if your main result is a 5 month study then should be very easy to find in the methods.*

We appreciate the opportunity to clarify. We have now reversed the order in which our samples are described, to reflect the importance of the ultrafines result. To rows 7-9 on p.16 we have added: “sampling included a seasonally similar period of 4.5 months – from January 20 to May 31, 2011 – of marked increase followed by decrease in ethanol prices. It is this sample that gives us heightened confidence in our submicron particle results.” P.6 of the earlier version stated “This was quite a seasonally homogeneous four-month period” – we have now standardized the text to “five months.” When providing a hypothetical example of where identification would fail, on p.21, row 10, also in Methods, we again consider the January to May 2011 sample. Instead of merely stating “Source: Specifications in Table II” in the caption of Fig. 1, we now cite the exact sample periods. To the caption of Fig. 3 we added: “...with sample period restricted to the summer/fall months of January to May 2011 and trend included as seasonality control.” In the previous version, sample periods were already stated clearly in the caption to Fig. 2 and in each table, e.g., Jan/2011 to May/2011 in column (3) of Table II. The main text also states that the results reported in these figures and Table II are based on the January to May 2011 sample, e.g., on p.8, row 7. Our estimates using the shorter sample of five months are robust to expanding the submicron sample, starting from October 2010 (Table IV describing regression estimates reported in Tables SVI and SVII).

The problem of using such a deterministic model is that exclusively depends on the initial variables considered, ignoring other processes and knowing “almost nothing” about the real mechanisms behind the increases or decreases of the nano-particles emissions (working as a black box).

8. The scope of our research is to provide an important and novel empirical benchmark – “innovative advances,” as per the Reviewer’s conclusion – for modelers of ambient air (who work on equally important and complementary research). It is based on actual, not hypothetical or modeled, changes in the fuel mix. This kind of variation happens rarely – it has not repeated itself since 2011 – and we were fortunate to have been ready, with instrumentation deployed – at least at one site – to exploit it. We again refer to Madronich (2014), as we did on p.2 of our earlier response letter, who described the “purely empirical approach” as “the gold standard for the type of analysis needed to evaluate the reliability of atmospheric chemistry models designed to simulate the effects of the transportation sector on air quality.”

9. Our study is not deterministic; it is stochastic in nature. It allows for econometric error, that is, the presence of unobserved determinants that the empiricist argues – by showing the evidence compiled from many sources – do not confound the estimated effect of interest. Effects are inferred by way of *confidence* intervals, not alleged “universal laws.” As we state on p.4, row 15, already in the earlier version, “The econometric approach requires that the analyst give careful consideration to whether remaining, unobservable determinants of nanoparticles might co-vary with the gasoline-ethanol mix, and the evidence suggests not.” We reiterate that the use of the word “suggest” rather than “prove” is indicative of the statistical scientific rigor that guides us.

One important role of empirics is to support and guide theoretical modeling of mechanisms, not to substitute for it. Our “identifying assumptions,” an ingredient to any econometric study, are clearly spelled out in expressions [2] to [4] in the Methods section: that, conditional on the included controls X , the gasoline-ethanol mix is orthogonal to unobserved determinants of particle levels (captured by the ε). As empirical scientists, one never observes all variables, but what is key here is that these unobserved factors do not move along with the gasoline-ethanol fuel mix. Of course, if there were some theoretically relevant unobserved factor that moved in step with the gasoline-ethanol mix, identification would fail. The following question then arises: Would the observed fluctuation in particle levels be due to the gasoline-ethanol mix, or to an unobserved, exogenous factor? (See the identification discussion on pp.21-22.) In the five-month window referenced in paragraph 7, the evidence is not supportive of the presence of such an unobserved factor shifting in one direction from January until March 2011 then reversing course until the end of the May. Even for the longer samples, we allow for trends to correct for gradual changes in other influences on particle levels, such as growth and compositional changes in the vehicle fleet (including technology, age).

You could have used meteorological parameters, vehicle fleet removal rates or data on local traffic changes instead and the results would had been completely different. It was the reason to cite at least other approaches and techniques.

10. We clarify that we do explicitly account for exactly this variation in meteorology, trends in the composition of vehicles/fuels, and traffic. These parameters are central ingredients of our econometric analysis, as we explain throughout.

[Editor's comment] *Clearly highlight exactly where in the manuscript this is.*

We discuss/describe variation in meteorology in: P4 R3, R14; P5 R12; P7 R14; P8 R3; P9 R1; P12 R1, R22; P15 R20; P16 subsection; P19 R19, R22;

We discuss/describe variation in local traffic in: P4 R4, R14; P5 R12; P8 R3; P12 R1, R22; P16 subsection; P19 R19, R23;

We discuss/describe variation in vehicle fleet in: (including longer term trends) P4 R13; P7 R12; P8 R5; P9 R3; P12 R1; P20 R5 (“secular changes in economic activity or in road and fleet composition”)

There are also some conceptual topics that must be clarified. In my understanding, you do not have trucks entering the university campus but bus lines are used for commuting. Buses are a key source of diesel emissions in São Paulo, as you mentioned, and maybe traffic conditions related to bus lines have changed during this period. Instead of using regional aggregate statistics on fuel consumption you could have used local transit data from companies (i.e. frequencies/headways of bus lines). I wonder how State's diesel prize policies could have affected the decrease of nano-particles at this sampling site. Once again, a redefinition of the research hypothesis and experiments should be done. The same for the other transport related fuel policies. Regarding O₃ formation, how can ridership affect O₃ formation changes? There must be another causes of this O₃ increments.

11. In response to the Reviewer's first-round comments, we examined diesel prices (in addition to quantities) for the city, not the state, of São Paulo. Again, this was done with a view to verify if diesel combustion might have shifted up and down in a way that could confound our analysis—the evidence does not support this hypothesis (Fig. S5).

12. We clarify that in the sensitivity analysis reported in Fig. S8, introduced in response to the Reviewer's first-round comments, we included diesel bus ridership as an additional control in the ozone regression model – itself meant as an illustration of the method as noted on p.7, row 16 – only for consistency with the other (particle) regression models. As can be seen comparing Fig. S8 to Fig. 1, whether we include a correction for diesel bus ridership or not makes no difference to the estimated effect of the gasoline-ethanol mix on ozone concentrations. Again, this is because diesel bus ridership did not covary with the gasoline-ethanol mix.

13. This week we contacted SPTrans (Prefeitura de São Paulo, Transportes) and the Prefeitura do Campus da Capital (PUSP-C). At this point, it is unclear whether historical bus transit data (lines, frequency, volume of commuters) specific to the USP campus are available. Instead, as a proxy for demand for bus services on campus, we note that enrollment at the University of Sao Paulo campus has remained broadly flat (or it exhibits a mild trend), as seen in the first table below. Thus, in the 4.5 month sample, for example, it is unlikely that bus activity would have shifted – in both directions – in step with the gasoline-ethanol fuel mix. We also note that, as pointed out above, the sampling site at the roof of the four-storey building is “without strong local sources... representative of the ambient pollution burden of the city... air masses arriving at the station should be well mixed” (Backman et al. 2012). Further, all our regression models already correct for local traffic flows (cars, bus, truck).

14. In response to the Reviewer, on p.17 we have added: “In the unlikely event that buses circulating on campus could exert a significant influence on the well-mixed urban air arriving at the measurement station, we obtained student enrollment at the USP (Armando Salles de Oliveira) campus. This should proxy for the demand for diesel bus services on the USP campus (or, rather, its variation) over the sample period. Enrollment over 2009-2013 has been very stable, for example, undergraduate enrollment varied by no more than 50 students around a mean enrollment of 7,451 students. Regular circulation of campus buses, or diesel vehicles anywhere, contribute to stable background levels of emissions that were unlikely to vary with the ethanol price fluctuations.”

**Table: Proxy for demand for bus services: Evolution of student enrollment and workforce. All campuses and Armando Salles de Oliveira campus only
Paints a picture of stability, if a mild trend**

	2008	2009	2010	2011	2012	2013
Undergraduate enrollment (all campuses)	55,863	56,998	57,300	57,902	58,303	
Graduate enrollment (all campuses)	25,495	25,591	26,568	27,795	28,498	
Enrollment:Staff ratio (all campuses)	14.4	14.4	14.3	14.4	14.8	
Undergraduate enrollment (Armando Salles de Oliveira campus only*)		7,465	7,484	7,418	7,439	7,451

Sources (accessed December 14, 2016):

Annual Report 2012, Table 1.01

https://uspdigital.usp.br/anuario/br/acervo/AnuarioUSP_2013.pdf

Annual Report 2015, Table 2.03

<https://uspdigital.usp.br/anuario/AnuarioControle#>

* Sum of the following units/departments (i.e., excluding campuses in Sao Carlos, Ribeirao, Sao Paulo Centro, Sao Paulo Dr Arnaldo, etc): EACH, ECA, EE, EEFE, EP, FAU, FCF, FE, FEA, FFLCH, FMVZ, FO, IAG, IB, ICB, IF, IGc, IME, IO, IP, IQ, IRI)

Reviewers' Comments:

Reviewer #2 (Remarks to the Author)

In this third round of observations and after a careful re-read of the manuscript, I still maintain the concerns regarding the main outcomes: trade-off between biofuel general use (particularly ethanol), O₃ increment and nano-particles (between 7 and 800nm & especially nucleation mode <50nm) transport emission decrease. The pseudo-empirical (called "purely empirical") and econometric approach used in the paper confirmed what was researched in other works related only to primary particle emissions not to secondary aerosols.

After analysing figure 2a, it seems quite difficult to believe in macro-economical terms that in such a short period (less than half a year), a 50-percentage point shift in the gasoline share in the flex fleet (from 30% to 80%) would have taken place. The energy consumption statistics from the national agency of petroleum ANP (collected in the supplementary material), during the period 2011-2012, showed that gasoline share steadily increased from 45% to 65% ("only" a 20% shift). How the artificial artifact of using a model to estimate gasoline/ethanol shares (ethanol-to-gasoline price ratio serving as an "instrumental variable" for the predicted gasoline shares) on a micro-economical basis can have influenced your main results? You affirmed that the positive relationship between the concentration in ultrafines and gasoline share is most significant during the morning rush hour. Are they using in São Paulo more gasoline during early morning commuting trips? Are those trips more elastic to fuel prizes than other types (leisure, study, etc.). Drawing conclusions based on an unique sampling plot and using the gasoline share micro-estimates to make conclusions for UFPs that are local source-dependent seems not to be a good proxy of the city's road transport paradigms and policy changes.

Although the authors reply the comments in the previous review to strength and justify in part their results - there was even an effort to include additional explanatory variables to their model (i.e. bus services/riderships but not vehicle technology changes and use) -, the manuscript tries to find a direct road emission source and air quality universal relationship based on UFP measurements in a "background" university campus spot. There are still unclear remarks in the methodological section (page 15 5-14): the paragraph is confusing since it seems you have bus stops nearby the sampling site inside the university campus ("vegetation on campus is limited compared to the dense vehicular traffic flows that surround it"). With these evident shortcomings, in part recognized now by the authors ("as with any empirical observational study, confidence in its findings can only grow as new samples, in space and time, become available, supported by the results from different approaches and analysis techniques"), it is up to the journal's editor to take the final decision (I do not need to review the paper again).

Reviewers' comments (in italics):

Reviewer #2 (Remarks to the Author):

In this third round of observations and after a carefull re-read of the manuscript, I still maintain the concerns regarding the main outcomes: trade-off between biofuel general use (particullarly ethanol), O3 increment and nano-particles (between 7 and 800nm & especially nucleation mode <50nm) transport emission decrease. The pseudo-empirical (called “purely empirical”) and econometric approach used in the paper confirmed what was researched in other works related only to primary particle emissions not to secondary aerosols.

We thank the Reviewer for his/her time reviewing our revised manuscript. We hope that our comments below and additional sensitivity analysis that we now include in the Supporting Information serve to allay remaining concerns. We emphasize that our study examines recent transitions by São Paulo’s consumers between gasoline and ethanol fuels (not “particularly” but specifically ethanol) and the effect on ambient particle pollution. The effect on ambient ozone is briefly added to illustrate the method. We respectfully disagree with the label “pseudo-empiricism.”

After analysing figure 2a, it seems quite difficult to believe in macro-economical terms that in such a short period (less than half a year), a 50-percentage point shift in the gasoline share in the flex fleet (from 30% to 80%) would have taken place.

Relative ethanol-gasoline prices shifted dramatically (Fig. S4). In principle, a flex-fuel vehicle can transition seamlessly between gasoline and ethanol, and the prices of both fuels at the pump are equalized – offering the same \$ per km traveled – when the ethanol-gasoline price ratio (\$/liter for ethanol E100 divided by \$/liter for gasoline E25) is around 0.70 (Supplementary Information (SI) Part C). Between October 2010 and May 2011, for example, the ethanol-gasoline price ratio shifted between 0.60 (ethanol very favorably priced relative to gasoline) and 0.85 (gasoline very favorably priced relative to ethanol), in both directions.

For perspective, moving the ratio from 0.60 to 0.85 is equivalent to a 42% relative ethanol price increase. Moving in the other direction implies a 29% relative ethanol price drop, or a 42% relative gasoline price increase. These are very large price shifts, widely reported in the contemporaneous media (e.g., radio, which motorists listen to while stuck in traffic). Prior to recent studies examining how consumers *actually* behaved at the pump (e.g., reference 30), the working assumption in the applied theoretical literature in environmental and energy economics (e.g., Holland et al. 2009) was that *all* flex-fuel vehicles would transition between fuels when the ethanol-gasoline ratio was at a tight range around 0.70. Thus, in the absence of consumer observation, a reader of Holland et al. (2009) would perhaps believe that a 50-percentage point shift in the gasoline share, in response to the large price shifts, was too low, rather than too high, as Reviewer #2 finds. (Please see the discussion in SI Part C, specifically on p.S8.)

The energy consumption statistics from the national agency of petroleum ANP (collected in the supplementary material), during the period 2011-2012, showed that gasoline share steadily increased from 45% to 65% (“only” a 20% shift).

To compare “apples to apples,” between October 2010 and May 2011: (i) the “blended gasoline share in the aggregate fleet of engines that are powered by either blended

gasoline (E20 or E25) or ethanol (E100)” shifted between 54% and 82% (see Fig. S4c, including the variable label), i.e., a 28-percentage point shift; and (ii) the “blended gasoline share in the flex-fuel light-vehicle fleet” shifted between 26% and 76% (Fig. S4a), i.e., a 50-percentage point shift.

As described in SI Part C (e.g., on p.S10), the evolution of these alternatively defined shares, share (i) and share (ii), is consistent. Share (i) is an *aggregate* share: aggregate across *vehicle types* (it includes consumers driving older gasoline-only vehicles and the predominantly gasoline-only motorcycles – not restricted to the fleet of flex-fuel vehicles as is share (ii)); aggregate in *time* (it varies only by month, not within the month as does share (ii)); and aggregate across *space* (it includes the entire state of São Paulo – including highway and rural markets rather than restricted to the São Paulo metropolis).

Moreover, in response to the comment “*only*” a 20% shift, it is important to note that a shift between 54% and 82% in the aggregate share is a 28-percentage point (not %) shift, i.e., moving from 54% to 82% is equivalent to a 51% gain ($82/54 - 1$), i.e., a gain of over one-half.

How the artificial artifact of using a model to estimate gasoline/ethanol shares (ethanol-to-gasoline price ratio serving as an “instrumental variable” for the predicted gasoline shares) on a micro-economical basis can have influenced your main results? It is to provide a sensitivity analysis to our baseline model based on ordinary least squares (OLS) that we provide estimates from “an alternative model based on two-stage least squares (2SLS), with the ethanol-to-gasoline price ratio serving as an ‘instrumental variable’ for the predicted gasoline share” (p.10, line 11). Both OLS and 2SLS (instrumental variables estimation) are widely established methods in statistics/econometrics/social sciences. For our OLS estimates, we use a bootstrap procedure to make a sampling correction – i.e., we conservatively increase the confidence bands – to account for the fact that, in a first step, the gasoline share is predicted from a demand system estimated from individual consumer choices and observed price variation at the pump. This is explained in Methods and SI Part C, where we provide relevant references. For example, on p.19 we state: “the reason we need a demand model to predict day-to-day fuel quantities from day-to-day fuel prices is that the former are not available for the São Paulo metropolis; otherwise, we would use the fuel quantity data directly in the second step and skip the first-step model.” The 2SLS estimates alternatively account for sampling error in the gasoline share prediction by use of an instrument (relative price) that moves with the predicted regressor (share) yet is uncorrelated with first-step sampling error.

Moreover, as we explain on p.23: “As a second alternative to using the gasoline share imputed for the São Paulo metropolis from an estimated consumer demand model, we use the gasoline share calculated from available aggregate monthly fuel quantity data reported by wholesalers for the entire state’s fleet.” (Also please see the text surrounding p.S10, line 14: “the alternative measure... based on aggregate monthly wholesale reported quantities for the entire state’s fleet, serves as a robustness check...”)

You affirmed that the positive relationship between the concentration in ultrafines and gasoline share is most significant during the morning rush hour. Are they using in São

Paulo more gasoline during early morning commuting trips? Are those trips more elastic to fuel prizes than other types (leisure, study, etc.). Drawing conclusions based on an unique sampling plot and using the gasoline share micro-estimates to make conclusions for UFPs that are local source-dependent seems not to be a good proxy of the city's road transport paradigms and policy changes.

We thank Reviewer #2 for his/her view but respectfully reiterate, to borrow from Backman et al. (2012) as we do in the manuscript (p.15), that the fourth-storey site in the University São Paulo campus is “representative of the ambient pollution burden of the city” and “ideal for tracking ambient aerosols, without strong local sources.”

Demonstrating that we did consider the Reviewer's concern on the possibility of local source dependence, spurred on by his/her round 2 comments, we obtained from São Paulo's public transportation authorities historical data on diesel buses transiting through campus (*actual*, not planned, trips). To p.15 we have now added: “records from São Paulo's public transportation authorities (SPTrans) over the submicron particle sampling period show only about one diesel bus transiting every two minutes within a horizontal 400 m radius of the site (Supplementary Figure S7).”

This new data is further described on pp. S12-S13. New figures reporting further sensitivity analysis, Figs. S13 and S14, show that: “the submicron particle results are robust to additionally controlling for observed variation in the frequency (always low) of public transit diesel buses passing within 400 m of the site at the time of sampling” (p.18).

Finally, with regard to “*the positive relationship between the concentration in ultrafines and gasoline share is most significant during the morning rush hour,*” it is possible that in the morning, after several hours of reduced anthropogenic activity, primary emissions from vehicles strongly impact the aerosol population, allowing for cleaner detection of an effect from gasoline versus ethanol use. In contrast, secondary processes including photochemistry that occur during the day and influence the nucleation sized aerosol concentration might make the effect less clearly observable. On p.9 and p.13, respectively, we state (already in the previous version):

“This relationship is not significant during the evening rush hour (though, given a CI of $-3,956$ to $+5,858$ cm^{-3} at 18:00, a positive effect cannot be statistically rejected either).”

“Whether subsequent atmospheric processing and/or secondary material formation are materially influenced by shifts in the fuel mix, and were not captured by our empirical model, is unknown and motivates further studies.”

Given current data and state of knowledge, we chose not to hypothesize whether and how much of the difference in significance across hours is due to atmospheric processing, or to consumer behavior, as the Reviewer alludes to.

(We also clarify that our research design does not rely on “*Are they using in São Paulo more gasoline during early morning commuting trips*” but rather whether consumers would, in response to price shifts, differentially switch between ethanol and gasoline according to their type (early morning commuters, leisure, study, as in the Reviewer's example).

Although the authors reply the comments in the previous review to strength and justify in part their results - there was even an effort to include additional explanatory

variables to their model (i.e. bus services/riderships but not vehicle technology changes and use) -, ...

We thank the Reviewer for acknowledging the effort that went into the previous review. We also take the opportunity to emphasize that changes to vehicle technology are gradual and are corrected for by including trends in the set of controls (see, e.g., p.4, line 13, p.7, line 14).

... the manuscript tries to find a direct road emission source and air quality universal relationship based on UFP measurements in a “background” university campus spot. The are still unclear remarks in the methodological section (page 15 5-14): the paragraph is confusing since it seems you have bus stops nearby the sampling site inside the university campus (“vegetation on campus is limited compared to the dense vehicular traffic flows that surround it”).

As stated above, to the paragraph the Reviewer references we now describe the new data on frequency of realized public diesel bus trips within 400 m of the sampling site. Any influence on the site of these passing buses (one every two minutes, in any direction), as well as of vegetation on campus, should pale in comparison to the influence of the dense vehicular traffic flows that surround the campus (the paragraph details the roads, representative of Sao Paulo’s road emissions). We also point out that we added the statement on vegetation in response to the Reviewer’s round 2 concern regarding “different sources even biological ones.”

With these evident shortcomings, in part recognize now by the authors (“as with any empirical observational study, confidence in its findings can only grow as new samples, in space and time, become available, supported by the results from different approaches and analysis techniques”), is up to the journal’s editor to take the final decision (I do not need to review the paper again).

References

Holland, S. P., J. E. Hughes and C. R. Knittel (2009). Greenhouse gas reductions under low carbon fuel standards? *American Economic Journal: Economic Policy* 1(1), 106-146